# Finite-Sample Maximum Likelihood Estimation of Location

**Shivam Gupta**
The University of Texas at Austin
shivamgupta@utexas.edu

**Jasper C.H. Lee**
University of Wisconsin–Madison
jasper.lee@wisc.edu

**Eric Price**
The University of Texas at Austin
ecprice@cs.utexas.edu

**Paul Valiant**
Purdue University
pvaliant@gmail.com

## Abstract

We consider 1-dimensional location estimation, where we estimate a parameter $\lambda$ from $n$ samples $\lambda + \eta_i$, with each $\eta_i$ drawn i.i.d. from a known distribution $f$. For fixed $f$ the maximum-likelihood estimate (MLE) is well-known to be optimal *in the limit* as $n \to \infty$: it is asymptotically normal with variance matching the Cramér-Rao lower bound of $\frac{1}{n\mathcal{I}}$, where $\mathcal{I}$ is the Fisher information of $f$. However, this bound does not hold for finite $n$, or when $f$ varies with $n$. We show for arbitrary $f$ and $n$ that one can recover a similar theory based on the Fisher information of a *smoothed* version of $f$, where the smoothing radius decays with $n$.

## 1  Introduction

We revisit a fundamental problem in statistics: consider a translation-invariant parametric model $\{f^\theta\}_{\theta \in \mathbb{R}}$ of distributions, where $f^\theta(x) = f(x - \theta)$. Suppose there is an arbitrarily chosen unknown true parameter $\lambda$, and we get i.i.d. samples from $f^\lambda$. The task is to accurately estimate $\lambda$ from the samples. This problem is known as *location parameter* estimation in the statistics literature.

Location estimation is a well-studied and general model, including as a special case the important setting of Gaussian mean estimation. In contrast to general mean estimation (where we want to estimate the mean of a distribution given minimal assumptions such as moment conditions), in location estimation we are given the shape of the distribution up to shift. This advantage lets us handle some distributions where mean estimation is impossible (e.g., the mean may not exist), and lets us aim for higher accuracy than is possible without knowing the distribution.

The classic theory of location estimation is asymptotic, see [vdV00] for a detailed background. On the algorithmic side, it is well-known that the maximum likelihood estimator (MLE) is asymptotically normal. Specifically, as the number of samples $n$ tends to infinity, the distribution of the MLE converges to a Gaussian centered at the true parameter with variance $1/(n\mathcal{I})$, where $\mathcal{I}$ is the *Fisher information* of the distribution $f$:

$$\mathcal{I} := \int \frac{(f'(x))^2}{f(x)} \, \mathrm{d}x = \mathop{\mathbb{E}}_{x \sim f}\left[ \left( \frac{\partial}{\partial x} \log f(x) \right)^2 \right]. \tag{1}$$

Conversely, the celebrated Cramér-Rao bound states that the variance of any unbiased estimator must be at least $1/(n\mathcal{I})$, meaning that the MLE has mean-squared error that is asymptotically at least as good as any unbiased estimator.

In the last few decades, motivated by an increasing dependence on data for high-stakes applications, the statistics and computer science communities have shifted focus towards *finite-sample* and *high*

*probability* theories: 1) asymptotic theories assume access to an infinite amount of data, and can in certain cases fail to predict the performance of an algorithm with only a finite number of samples—see the next section for bad examples for the MLE— 2) in high-stakes applications where failure can be catastrophic, it is crucial for predictions to hold except with exponentially small probability. Yet, classic results bounding the variance or mean-squared error of estimators, such as the Cramér-Rao bound, do not readily translate to (tight) high probability bounds.

The goal of this paper is to establish a finite-sample and high probability theory for the location estimation problem, in both the algorithmic bound and the estimation lower bound. Our algorithmic theory includes a simple yet crucial modification of perturbing samples by Gaussian noise—corresponding to drawing samples from a Gaussian-smoothed version of the underlying distribution—before performing MLE. We show that this smoothed MLE has finite-$n$ high-probability performance analogous to the Gaussian tail in the classic asymptotic theory, but replacing the usual Fisher information with a *smoothed Fisher information*. The amount of smoothing required decreases with $n$.

Complementing our upper bound result, we prove a high probability version of the Cramér-Rao bound for Gaussian-smoothed distributions, showing that for these instances our sub-Gaussian accuracy bound, with variance determined by the Fisher information, is optimal to within a $1 + o(1)$ factor.

## 1.1 Obstacles to a Finite Sample Theory

Before discussing our results in detail, we examine two simple distributions where the asymptotic theory predictions for the MLE do not hold in finite samples. The first example highlights an information-theoretic barrier, that no algorithm can attain the performance predicted by the Gaussian with variance $1/\mathcal{I}$. The second example, on the other hand, demonstrates how the MLE can be "tricked" by the distribution, and how an algorithmic remedy is needed to improve its accuracy.

**Gaussian with sawtooth noise**   Our first example takes a standard Gaussian, and adds a fine-grained sawtooth perturbation to it over a bounded region at the center of the Gaussian, as shown in Figure 1a. The fine-grained sawtooth has slope either $+\Delta$ or $-\Delta$, alternating over "teeth" of width $w$, for $\Delta \gg 1$ and $w \ll 1/\Delta$. This sawtooth perturbation barely changes the pdf, but significantly changes its derivatives, so the Fisher information $\int (f')^2/f$ grows from 1 to $\Theta(\Delta^2)$.

Essentially, the sawtooth perturbation makes the distribution easier to align *within* a tooth, but not *across* teeth. The asymptotic analysis reflects that: for $n \gg 1/w^2$ where we can align the teeth correctly, the MLE is much more accurate on the perturbed distribution. But for $n \ll 1/w^2$, no algorithm can do better on the perturbed distribution than on a regular Gaussian.[1] Thus, the normalized estimation accuracy depends on $n$: for large enough $n$, it has variance $\frac{\Theta(1)}{\Delta^2 n}$, but for smaller $n$ it has variance $\frac{1}{n}$. Our finite sample theory should reflect this.

A formal statement of the above reasoning is as follows, with a proof sketch in Appendix A.

**Proposition 1.1.** *The Gaussian+sawtooth model, with "teeth" of width $w$ and slope $\Delta$, has Fisher information $\Delta^2 \gg 1$ but no location estimator can have error $o(1/\sqrt{n})$ with constant probability over $n$ samples, unless $n > 0.01/w^2$.*

*This holds for arbitrarily large $\Delta$ and small $w$. By contrast, the asymptotic theory predicts error $O(1/(\Delta\sqrt{n}))$, which only holds for $n \gtrsim 1/w^2$.*

**Gaussian with a Dirac $\delta$ spike**   Our next example adds an even simpler noise to the standard Gaussian: a Dirac $\delta$ spike with minimal mass $\varepsilon$, placed sufficiently far away from the Gaussian mean 0 (Figure 1b). This distribution has infinite Fisher information, reflecting that for $n \gg 1/\varepsilon$, we will probably see the spike multiple times, identify it, and get zero estimation error. But for $n < 1/\varepsilon$, we probably will not see the spike, so our error bound will not reflect the overall Fisher information.

Moreover, the MLE performs remarkably badly when $n \ll 1/\varepsilon$. Most likely the Dirac $\delta$ is not sampled, yet the Dirac $\delta$ has infinite density, and so the MLE will match a Gaussian sample to the spike to get a maximum likelihood solution. As the spike is placed far from the true mean, this leads to much higher error than (say) the empirical mean.

The reasoning in the above two paragraphs can be formally summarized as follows.

---

[1]This can be shown by the KL divergence from shifting an integer number of teeth of distance about $1/\sqrt{n}$.

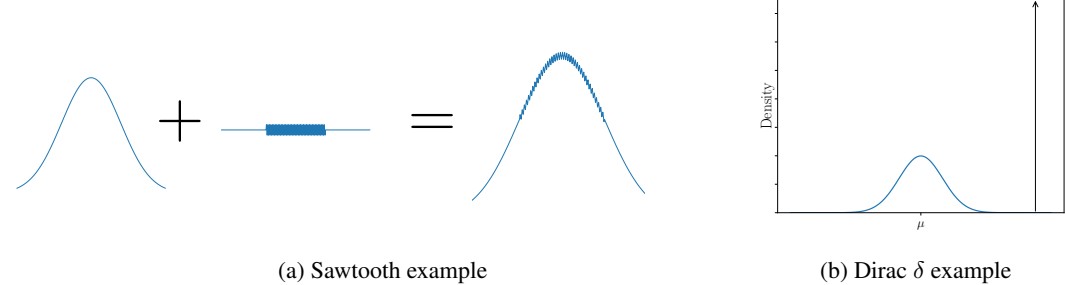

(a) Sawtooth example

(b) Dirac $\delta$ example

Figure 1: Bad examples for MLE

**Proposition 1.2.** *Consider the spiked Gaussian example $(1 - \tau)\mathcal{N}(0, 1) + \tau\delta_T$ where $\delta_T$ is a Dirac $\delta$ at location $x = T$. The Fisher information is infinite, but for $n < 1/(100\tau)$, every estimation algorithm has median error at least $0.6/\sqrt{n}$. Moreover, with $98\%$ probability the MLE has error at least $T - O(\sqrt{\log n})$, which can be arbitrarily large.*

We also note that, while we present the hardness result in terms of mixing a Gaussian with a Dirac $\delta$ spike for conceptual simplicity, a milder but similar effect occurs even when the spike is narrow but not infinite.

Fortunately, there is a simple modification to the MLE that solves this issue almost completely: we add a small amount of Gaussian noise to the distribution. This means we convolve the PDF with a small Gaussian, and add independent Gaussian noise to each individual sample to effectively draw from the convolved PDF. The crucial effect of this Gaussian smoothing is that the density of the Dirac $\delta$ will be reduced from infinity down to some constant, and hence MLE will no longer "fit it at any cost". On the other hand, the smoothing increases the variance of the distribution and decreases the Fisher information. This raises the question of determining the *best* amount of smoothing, which we address in this paper.

## 1.2 Our Results and Approach

Based on the discussion of the spiked Gaussian model in the previous section, we propose a finite sample theory for MLE that uses Gaussian smoothing. As we described, the algorithmic approach is simple: we perturb all the samples by independent Gaussian noise of variance $r^2$, and convolve the known model $f$ by the same Gaussian to yield the model $f_r$, before performing MLE.

We state below the basic MLE algorithm (Algorithm 1) in this paper, which adopts the above approach. Algorithm 1 is a *local* algorithm, in that it assumes as input an initial uncertainty region guaranteed to contain the true parameter $\lambda$, and performs MLE only over this domain. Furthermore, Algorithm 1 attempts only to find a local optimum in the likelihood: it computes the derivative of the log likelihood function, also known as the *score* function, and returns any root of the score function.

---

**Algorithm 1** Local MLE for known parametric model

---

**Input Parameters:** Description of distribution $f$, smoothing parameter $r$, samples $x_1, \ldots, x_n \overset{i.i.d.}{\sim} f^\lambda$, uncertainty region $[\ell, u]$ containing the unknown $\lambda$

1. Let $s_r(\hat{\lambda})$ be the score function of $f_r$, the $r$-smoothed version of $f$.

2. For each sample $x_i$, compute a perturbed sample $x_i' = x_i + \mathcal{N}(0, r^2)$ where all the Gaussian noise are drawn independently across all the samples.

3. Compute $\hat{\lambda}$ that is a root of the empirical score function $\hat{s}(\hat{\lambda}) = \frac{1}{n}\sum_{i=1}^{n} s_r(x_i' - \hat{\lambda})$ inside the domain $[\ell, u]$. A root should exist and picking any root is sufficient.

4. Return $\hat{\lambda}$.

---

Theorem 1.3 states our guarantees on Algorithm 1. Locally around the true parameter—that is, within $r/2$ for the $r$-smoothed distribution—any root of the score function (i.e., local optimum of likelihood) must be very close to the true parameter $\lambda$ with high probability over the $n$ samples. In particular, the

estimation error is within a $1 + o(1)$ factor of the Gaussian deviation with variance $\frac{1}{n\mathcal{I}_r}$ and failure probability $\delta$, where $\mathcal{I}_r$ is the Fisher information of $f_r$.

**Theorem 1.3** (Local Convergence). *Suppose we have a known model $f_r$ that is the result of $r$-smoothing, with Fisher information $\mathcal{I}_r$, and a given width parameter $\varepsilon_{\max}$ and failure probability $\delta$. Further suppose that $r$ satisfies $r \geq 2\varepsilon_{\max}$ and there is a parameter $\gamma$, where $\gamma$ is larger than some universal constant, such that 1) $r^2\sqrt{\mathcal{I}_r} \geq \gamma\varepsilon_{\max}$, 2) $(\log \frac{1}{\delta})/n \leq \frac{1}{\gamma^2}$ and 3) $\log 1/(r\sqrt{\mathcal{I}_r}) \leq \frac{1}{\gamma}\log\frac{1}{\delta}/\log\log\frac{1}{\delta}$.*

*Then, with probability at least $1 - \delta$, for all $\varepsilon \in \left((1 + O(\frac{1}{\gamma}))\sqrt{\frac{2\log\frac{1}{\delta}}{n\mathcal{I}_r}}, \varepsilon_{\max}\right]$, $\hat{s}(\lambda - \varepsilon)$ is strictly negative and $\hat{s}(\lambda + \varepsilon)$ is strictly positive.*

In the scenario where we do have an initial uncertainty region for the true parameter $\lambda$, we would use Theorem 1.3 to compute the minimal smoothing amount $r$ satisfying the assumptions in the theorem, then use Algorithm 1 with this parameter $r$, to obtain an accurate estimate of $\lambda$.

In general, however, we may not have an initial uncertainty region for $\lambda$. In Section 5, we present Algorithm 2, a global two-stage MLE algorithm, which first infers an initial uncertainty region by using quantile information from the distribution $f$ before invoking Algorithm 1, the local MLE algorithm. The guarantees of Algorithm 2 are summarized here.

**Theorem 1.4** (Global MLE guarantees, informal version of Theorem 5.1). *Given a model $f$, let the $r$-smoothed Fisher information of a distribution $f$ be $\mathcal{I}_r$, and let $\mathrm{IQR}$ be the interquartile range of $f$. When $n \gg \log\frac{1}{\delta} \gg 1$, there exists an $r^* = o(\mathrm{IQR})$ such that, with probability at least $1 - \delta$, the output $\hat{\lambda}$ of Algorithm 2 satisfies*

$$|\hat{\lambda} - \lambda| \leq (1 + o(1))\sqrt{\frac{2\log\frac{1}{\delta}}{nI_{r^*}}}$$

In addition to the theoretical framework, Section 7 gives experimental evidence demonstrating that $r$-smoothed Fisher information does capture the empirical performance of (smoothed) MLE.

We also prove new estimation lower bounds for the location estimation problem for $r$-smoothed distributions. The lower bound statement below (Theorem 1.5) shows that the estimation error $(1 + o(1))\sqrt{\frac{2\log\frac{1}{\delta}}{nI_r}}$ is optimal to within a $1 + o(1)$ factor.

**Theorem 1.5.** *Suppose $f_r$ is an $r$-smoothed distribution with Fisher information $\mathcal{I}_r$. Given failure probability $\delta$ and sample size $n$, no algorithm can distinguish $f_r$ and $f_r^{2\varepsilon}$ with probability $1 - \delta$, where $\varepsilon = (1 - o(1))\sqrt{2\log\frac{1}{\delta}/(n\mathcal{I}_r)}$. Here, the $o(1)$ term tends to 0 as $\delta \to 0$ and $\log\frac{1}{\delta}/n \to 0$, for a fixed $r^2\mathcal{I}_r$.*

This lower bound is the standard "two-point" statement that, with $n$ samples, it is statistically impossible to distinguish between the distributions $f$ and $f$ shifted by a small error (in the $x$-axis) with probability $1 - \delta$. Even though there are known standard inequalities on distribution distances and divergences for proving lower bounds of this form, the technical challenge is that they generally yield estimation lower bounds that are only tight to within constant factors, instead of the $1 + o(1)$ tightness we desire. This paper presents new analysis to derive a $1 + o(1)$-tight lower bound, which may be of independent interest.

### 1.3 Notations

We denote the shift-invariant model we consider by the distribution $f$, and the distribution with parameter $\lambda$ by $f^{\lambda}(x) = f(x - \lambda)$. Denote by $Z_r$ the Gaussian with mean 0 and variance $r^2$. The $r$-smoothed model for $f$ is denoted by $f_r$ (and similarly, for parameter $\lambda$, $f_r^{\lambda}$) which is distributed as $Y = Z_r + X$ where $X \leftarrow f$ independently from the Gaussian perturbation $Z_r$.

The log-likelihood function of $f$ is denoted by $l = \log f$. The *score* function is the derivative of $l$, denoted by $s = l' = f'/f$. We use the notation $s_r$ to denote the score function of $f_r$. The Fisher Information of $f$ is denoted by $I = \mathbb{E}_{x \leftarrow f}[s^2(x)]$. Similarly, the Fisher Information of $f_r$ is denoted by $I_r$.

## 2 Related Work

Location estimation and MLE in general has been extensively studied under the lens of asymptotic statistics. See [vdV00] for an in-depth treatment. The MLE has also been studied under the finite-sample setting [Spo11, Pin17, VdG00, Mia10], but these prior works impose restrictive regularity conditions and also loses (at least) multiplicative constants in the estimation accuracy. Specifically, [Spo11] assumes that the parametric family $f^\theta$ satisfies $D_{\mathrm{KL}}(f^\theta \parallel f^{\theta'}) \geq \Omega(\mathcal{I}|\theta - \theta'|^2)$ for every pair of parameters $\theta, \theta'$, where $\mathcal{I}$ is the Fisher information of the parametric family. This is a *global* distance property that does not always hold in our general setting, for example in the Gaussian with sawtooth example in Section 1.1, where $\mathcal{I}$ can be arbitrarily large yet the density is arbitrarily close to the standard Gaussian. Our work handles the lack of global distance property by instead proposing a two-stage final algorithm (Algorithm 2 in Section 5). [Pin17] also assumes a global distance property, similar to [Spo11], but in Hellinger distance instead of KL divergence. [VdG00] again assumes a somewhat similar global distance property, and further characterizes convergence of the recovered distribution in Hellinger distance to the true distribution, instead of directly bounding estimation error in the parameter space. [Mia10] on the other hand requires Lipschitzness in the score, whereas for Gaussian-smoothed distributions we only have 1-sided Lipschitzness (Lemma B.2). Moreover, the [Mia10] results quantify the error not in terms of the Fisher information (the variance of the score), but in terms of the maximum magnitude of the score.

In contrast to these prior works, our work modifies the MLE to include smoothing. Without smoothing, these prior works do not apply even in simple cases like a standard Laplace distribution: [Spo11] because the KL divergence between shifted versions of the Laplace distribution is not locally quadratic in the shift-parameter space, and [Pin17] because of this reason and their further assumption that the score function is smooth, which is not true at the origin for the Laplace distribution. On the other hand, by introducing smoothing to the MLE, we can take an *arbitrary* distribution and extract a meaningful finite-sample guarantee, which is tight to within $1 + o(1)$ factors conditioned on such smoothing.

There has also been a flurry of recent interest in the related mean estimation problem, in the finite-sample and high-probability setting. Recall that mean estimation does not assume knowledge of the shape of the distribution, but instead imposes mild moment conditions, for example the finiteness of the variance. Catoni [Cat12] initiated a line of work studying the statistical limits of univariate mean estimation to within a $1 + o(1)$ factor, ending recently with the work of Lee and Valiant [LV22], which proposed and analyzed an estimator with accuracy optimal to within a $1 + o(1)$ factor for all distributions with finite variance. See also the recent work of Minsker [Min22] for an alternative solution.

Beyond the differences in assumptions, the main distinction between location and mean estimation lies in their statistical limits. In mean estimation, the optimal accuracy is captured by the variance of the underlying distribution, scaling linearly with the standard deviation. On the other hand, the classic asymptotic theory suggests that the Fisher information captures the optimal accuracy for location estimation, scaling with the reciprocal of the square root of the Fisher information. It is a well-known fact that the Fisher information is always lower bounded by the reciprocal of the variance [SV11], which shows that location estimation is always easier than mean estimation in the infinite-sample regime. In this work, we refine this understanding, showing that in finite samples, the optimal accuracy for location estimation is instead given by the $r$-smoothed Fisher information in place of the unsmoothed Fisher information.

## 3 Tails and boundedness of $r$-smoothed score and Fisher information

Recall that given a distribution $f$, its $r$-smoothed version $f_r$ is distributed as $Y = X + Z_r$ where $X \sim f$ and $Z_r \sim \mathcal{N}(0, r^2)$ and $X, Z_r$ are independent.

Both our algorithmic and lower bound theories are centered around $r$-smoothed distributions. Therefore, we state here basic concentration and boundedness properties of $r$-smoothed score function and Fisher information, which we use in the rest of the paper. We prove all these lemmas in Appendix B.

First, we show that the $r$-smoothed Fisher information $\mathcal{I}_r$ is upper bounded by $1/r^2$ and can be lower bounded using the interquartile range of $f$.

**Lemma 3.1.** *Let $\mathcal{I}_r$ be the Fisher information of an $r$-smoothed distribution $f_r$. Then, $\mathcal{I}_r \leq 1/r^2$.*

**Lemma 3.2.** *Let $\mathcal{I}_r$ be the Fisher information for $f_r$, the $r$-smoothed version of distribution $f$. Let IQR be the interquartile range of $f$. Then, $\mathcal{I}_r \gtrsim 1/(\text{IQR} + r)^2$. Here, the hidden constant is a universal one independent of the distribution $f$ and independent of $r$.*

Next, we show that, fixing a point close to the true parameter $\lambda$, the empirical score function evaluated at that point will concentrate around its expectation for smoothed distributions.

**Corollary 3.3.** *Let $f$ be an arbitrary distribution and let $f_r$ be the $r$-smoothed version of $f$. That is, $f_r(x) = \mathbb{E}_{y \leftarrow f}[\frac{1}{\sqrt{2\pi r^2}} e^{-\frac{(x-y)^2}{2r^2}}]$. Consider the parametric family of distributions $f_r^\lambda(x) = f_r(x - \lambda)$. Suppose we take $n$ i.i.d. samples $y_1, \ldots, y_n \leftarrow f_r^\lambda$, and consider the empirical score function $\hat{s}$ mapping a candidate parameter $\hat{\lambda}$ to $\frac{1}{n} \sum_i s_r(y_i - \hat{\lambda})$, where $s_r$ is the score function of $f_r$.*

*Then, for any $|\varepsilon| \leq r/2$,*

$$
\Pr_{y_i \overset{i.i.d.}{\sim} f_r^\lambda} \left( |\hat{s}(\lambda + \varepsilon) - \mathbb{E}_{x \leftarrow f_r}[s(x - \varepsilon)]| \geq \sqrt{\frac{2 \max(\mathbb{E}_x[s_r^2(x - \varepsilon)], \mathcal{I}_r) \log \frac{2}{\delta}}{n}} + \frac{15 \log \frac{2}{\delta}}{nr} \right) \leq \delta
$$

# 4 A Finite Sample Analysis of $r$-smoothed Local MLE

In this section, we analyze Algorithm 1, which is our version of local MLE with $r$-smoothing applied. Algorithm 1 takes an initial uncertainty region that the true parameter is guaranteed to lie in, and uses the model and the initial interval to refine the estimate to high accuracy. We first present a simpler and easier-to-interpret version of our result, Theorem 1.3, which we stated in Section 1.2.

Recall that Algorithm 1 computes the empirical score function, and returns any of its roots. The theorem thus states that, with high probability, for any point $\lambda + \varepsilon$ with $|\varepsilon|$ too large, the empirical score function must be non-zero and thus $\lambda + \varepsilon$ will not returned as the estimate. More precisely, given an initial interval of length $\varepsilon_{\max}$ as well as the failure probability $\delta$, the theorem assumes that the smoothing parameter $r$ is sufficiently large (conditions 1 and 3 in the theorem) and that the sample size $n$ is sufficiently large, and guarantees an estimation error that is within a $1 + o(1)$ factor of the error predicted by the Gaussian with variance $1/\mathcal{I}_r$, where $\mathcal{I}_r$ is the Fisher information of $f_r$.

**Theorem 1.3** (Local Convergence). *Suppose we have a known model $f_r$ that is the result of $r$-smoothing, with Fisher information $\mathcal{I}_r$, and a given width parameter $\varepsilon_{\max}$ and failure probability $\delta$. Further suppose that $r$ satisfies $r \geq 2\varepsilon_{\max}$ and there is a parameter $\gamma$, where $\gamma$ is larger than some universal constant, such that 1) $r^2 \sqrt{\mathcal{I}_r} \geq \gamma \varepsilon_{\max}$, 2) $(\log \frac{1}{\delta})/n \leq \frac{1}{\gamma^2}$ and 3) $\log 1/(r\sqrt{\mathcal{I}_r}) \leq \frac{1}{\gamma} \log \frac{1}{\delta} / \log \log \frac{1}{\delta}$.*

*Then, with probability at least $1 - \delta$, for all $\varepsilon \in \left( (1 + O(\frac{1}{\gamma})) \sqrt{\frac{2 \log \frac{1}{\delta}}{n \mathcal{I}_r}}, \varepsilon_{\max} \right]$, $\hat{s}(\lambda - \varepsilon)$ is strictly negative and $\hat{s}(\lambda + \varepsilon)$ is strictly positive.*

Theorem 1.3 follows from Theorem 4.1 below, which makes the "$o(1)$" term (the $O(1/\gamma)$ term) in the theorem explicit. Assumptions 2 and 3 in the theorem statement essentially bounds various multiplicative terms in the estimation error and makes sure that they are "$1 + o(1)$" terms. In Appendix C, we give the formal proof of Theorem 1.3 using Theorem 4.1.

**Theorem 4.1.** *Suppose we have a known model $f_r$ that is the result of $r$-smoothing, and a given parameter $\varepsilon_{\max}$. Let $\beta$ and $\eta$ be the hidden multiplicative constants in Lemmas C.2 and C.3. Further suppose that $r$ satisfies $r \geq 2\varepsilon_{\max}$ and $r^2 \sqrt{\mathcal{I}_r} \geq \gamma \varepsilon_{\max}$ for some parameter $\gamma \geq \beta$.*

*Now define the notation $\rho_r$ by*

$$
1 + \rho_r = \sqrt{1 + \frac{\eta \sqrt{\varepsilon}}{\gamma}} + \frac{15}{2\sqrt{\gamma}} \left( \frac{2 \log \frac{4 \log \frac{1}{\delta}}{r^2 \mathcal{I}_r (1 - \frac{\beta}{\gamma})\delta}}{n} \right)^{\frac{1}{4}}
$$

*Then, for sufficiently small* $\delta > 0$, *with probability at least* $1 - \delta$, *for all* $\varepsilon \in$

$$\left( (1 + \tfrac{1}{\log \frac{1}{\delta}}) \tfrac{1+\rho_r}{1-\frac{\beta}{\gamma}} \sqrt{1 + \frac{\log \frac{4 \log \frac{1}{\delta}}{r^2 \mathcal{I}_r (1 - \frac{\beta}{\gamma})}}{\log \frac{1}{\delta}}} \sqrt{\frac{2 \log \frac{1}{\delta}}{n \mathcal{I}_r}}, \varepsilon_{\max} \right], \hat{s}(\lambda - \varepsilon) < 0 \ and \ \hat{s}(\lambda + \varepsilon) > 0.$$

To prove Theorem 4.1, it suffices to show the following lemma. The theorem follows directly by reparameterizing $\delta$ and choosing $\xi$ to be $1/\log \frac{1}{\delta}$.

**Lemma 4.2.** *Suppose we have a known model $f_r$ that is the result of $r$-smoothing with Fisher information $\mathcal{I}_r$, and a given parameter $\varepsilon_{\max}$. Let $\beta$ and $\eta$ be the hidden multiplicative constants in Lemmas C.2 and C.3. Further suppose that $r$ satisfies $r \geq 2\varepsilon_{\max}$ and $r^2 \sqrt{\mathcal{I}_r} \geq \gamma \varepsilon_{\max}$ for some parameter $\gamma \geq \beta$. Also define the notation $\tilde{\rho}$ (a "$o(1)$" term) by*

$$1 + \tilde{\rho} = \sqrt{1 + \frac{\eta \sqrt{\varepsilon}}{\gamma}} + \frac{15}{2\sqrt{\gamma}} \left( \frac{2 \log \frac{1}{\delta}}{n} \right)^{\frac{1}{4}}$$

*Then, for every $\xi \ll 1$, with probability at least $1 - \delta \cdot \frac{2}{\xi r^2 \mathcal{I}_r (1 - \frac{\beta}{\gamma})(1-\delta)}$, for all $\varepsilon \in$*

$$\left( (1 + \xi) \tfrac{1+\tilde{\rho}}{1-\frac{\beta}{\gamma}} \sqrt{\frac{2 \log \frac{1}{\delta}}{n \mathcal{I}_r}}, \varepsilon_{\max} \right], \hat{s}(\lambda - \varepsilon) \ is \ strictly \ negative \ and \ \hat{s}(\lambda + \varepsilon) \ is \ strictly \ positive.$$

We prove Lemma 4.2 in Appendix C, and here we give a proof sketch.

*Proof sketch for Lemma 4.2.* First, recall that Corollary 3.3 from Section 3 shows that fixing a candidate input value $\lambda + \varepsilon$ for some small $\varepsilon$, the value of the empirical score function at $\lambda + \varepsilon$ is well-concentrated around its expectation. In Lemmas C.2 and C.3, we calculate and bound the expectation and second moment of the empirical score function at $\lambda + \varepsilon$ for all sufficiently small $\varepsilon$. This allows us to derive tail bounds for the empirical score function at each point $\lambda + \varepsilon$, to show that it is bounded away from 0. Next, we need to show that with high probability, the empirical score function is *simultaneously* bounded away from 0 for all $\varepsilon$ with magnitude greater than the desired estimation accuracy. We achieve this via a straightforward net argument, crucially utilizing the fact that the expectation of the empirical score function is bounded away from 0 by an essentially linear function in $\varepsilon$, and that the variance is essentially constant in $\varepsilon$. This means that the probability for the empirical score function at $\lambda + \varepsilon$ to hit 0 is decreasing exponentially in $\varepsilon$, which allows us to complete the net argument. $\square$

# 5 Global Two-Stage MLE Algorithm

Algorithm 1, which we stated in the introduction and analyzed in Section 4, is a *local* algorithm that assumes we have knowledge of a non-trivially small uncertainty region containing the true parameter $\lambda$. The smoothing parameter $r$ can then be computed from the assumptions of Theorems 4.1 or 1.3, and we run Algorithm 1 to obtain an accurate estimate of the true parameter $\lambda$, with accuracy predicted by the $r$-smoothed Fisher information $\mathcal{I}_r$.

However, in general, we might not have a-priori knowledge of where the true parameter $\lambda$ lies. In this section, we propose a *global* maximum likelihood algorithm (Algorithm 2) which first estimates a preliminary interval containing $\lambda$, before choosing the smoothing parameter $r^*$ using an *easily calculable* expression that is $o(1)$ times smaller than the interquartile range of the distribution, and finally applies the local MLE algorithm (Algorithm 1) to obtain a final estimate. Theorem 5.1 states that the accuracy of Algorithm 2 is always within a $1 + o(1)$ times the accuracy predicted by the $r^*$-smoothed Fisher information $\mathcal{I}_{r^*}$.

**Theorem 5.1** (Global MLE Theorem). *Given a model $f$, let the $r$-smoothed Fisher information of a distribution $f$ be $\mathcal{I}_r$, and let IQR be the interquartile range of $f$. Fix the failure probability be $\delta$.*

*Choose $r^* = \Omega(\max((\frac{\log \frac{1}{\delta}}{n})^{1/8}, 2^{-O(\sqrt{\log \frac{1}{\delta}})}))$IQR. Then, with probability at least $1 - \delta$, the output $\hat{\lambda}$ of Algorithm 2 satisfies*

$$|\hat{\lambda} - \lambda| \leq \left( 1 + O\left( \frac{\log \frac{1}{\delta}}{n} \right)^{\frac{1}{4}} + O\left( \frac{1}{\sqrt{\log \frac{1}{\delta}}} \right) \right) \sqrt{\frac{2 \log \frac{1}{\delta}}{n \mathcal{I}_{r^*}}}$$

---
**Algorithm 2** Global MLE for known parametric model
---
**Input Parameters:** Failure probability $\delta$, description of distribution $f$, $n$ i.i.d. samples drawn from $f^\lambda$ for some unknown $\lambda$

1. Compute an $\alpha \in [\sqrt{2\log\frac{4}{\delta}/n}, 1 - \sqrt{2\log\frac{4}{\delta}/n}]$ such that the interval defined by the $\alpha - \sqrt{2\log\frac{4}{\delta}/n}$ and $\alpha + \sqrt{2\log\frac{4}{\delta}/n}$ quantiles of $f$ is the smallest.

2. By standard Chernoff bounds, with probability at least $1 - \frac{\delta}{2}$, the sample $\alpha$-quantile $x_\alpha$ will be such that $x_\alpha - \lambda$ is within the $\alpha - \sqrt{2\log\frac{4}{\delta}/n}$ and $\alpha + \sqrt{2\log\frac{4}{\delta}/n}$ quantiles of $f$. Based on this, compute an initial confidence interval $[\ell, u]$ for $\lambda$.

3. Let $r^* = \Omega(\max((\frac{\log\frac{1}{\delta}}{n})^{1/8}, 2^{-O(\sqrt{\log\frac{1}{\delta}})}))\mathrm{IQR}$.

4. Run Algorithm 1 on the interval $[\ell, u]$, using $r^*$-smoothing and failure probability $\delta/2$, returning the final estimate $\hat{\lambda}$.
---

*Proof.* The total failure probability of the steps is at most $\delta$. Thus, in this proof we condition on the success of Algorithm 2 in all probabilistic steps.

By the minimality condition in the definition of $\alpha$, the length $\varepsilon_{\max}$ of the interval $[\ell, u]$ from Step 2 is at most $O(\sqrt{\log\frac{1}{\delta}/n})\mathrm{IQR}$.

Further, recall by Lemma 3.2 that $\mathcal{I}_r \geq \Omega(1/(\mathrm{IQR} + r)^2)$. Picking $r^* = \Omega(\max((\frac{\log\frac{1}{\delta}}{n})^{1/8}, 2^{-O(\sqrt{\log\frac{1}{\delta}})}))\mathrm{IQR}$ and $\gamma_1 = O(\frac{n}{\log\frac{1}{\delta}})^{1/4}$ and $\gamma_2 = O(\sqrt{\log\frac{1}{\delta}})$, we check that the following conditions are satisfied:

1. $(r^*)^2\sqrt{\mathcal{I}_{r^*}} \geq (r^*)^2/(\mathrm{IQR} + r^*) \geq \Omega(\frac{\log\frac{1}{\delta}}{n})^{1/4}\mathrm{IQR} = \gamma_1\varepsilon_{\max}$.

2. $\log\frac{1}{\delta}/n \leq O(\sqrt{\log\frac{1}{\delta}/n}) \leq 1/\gamma_1^2$.

3. $\log 1/(r^*\sqrt{\mathcal{I}_{r^*}}) \leq O(\log 2^{O(\sqrt{\log\frac{1}{\delta}})}) = O(\frac{1}{\gamma_2}\log\frac{1}{\delta})$.

Further note that $\log\log\frac{1}{\delta}/\log\frac{1}{\delta} \leq 1/\sqrt{\log\frac{1}{\delta}} = O(1/\gamma_2)$.

Thus, using Theorem 1.3, Step 4 returns an estimate $\hat{\lambda}$ satisfying

$$|\hat{\lambda} - \lambda| \leq \left(1 + O\left(\frac{1}{\gamma_1}\right) + O\left(\frac{1}{\gamma_2}\right)\right)\sqrt{\frac{2\log\frac{1}{\delta}}{n\mathcal{I}_{r^*}}}$$

which is equivalent to the theorem statement. $\qquad\square$

# 6 High Probability Cramér-Rao Bound

Complementing our algorithmic results, we also give new results on *lower bounding* the estimation error in the location parameter model. The celebrated Cramér-Rao bound lower bounds the variance of estimators, which does not readily translate to (tight) lower bounds on the distribution tail of the estimation error. In this section, we show that it is possible to derive a high probability version of the Cramér-Rao lower bound for $r$-smoothed distributions, where, given a failure probability $\delta$, we lower bound the estimation error to within a $1 + o(1)$-factor of the error predicted by the asymptotic normality of the standard maximum likelihood algorithm, namely the Gaussian with the true parameter as the mean, and variance $1/(n\mathcal{I}_r)$ for estimation using $n$ samples.

**Theorem 1.5.** *Suppose $f_r$ is an $r$-smoothed distribution with Fisher information $\mathcal{I}_r$. Given failure probability $\delta$ and sample size $n$, no algorithm can distinguish $f_r$ and $f_r^{2\varepsilon}$ with probability $1 - \delta$,*

where $\varepsilon = (1 - o(1))\sqrt{2\log\frac{1}{\delta}/(n\mathcal{I}_r)}$. *Here, the $o(1)$ term tends to 0 as $\delta \to 0$ and $\log\frac{1}{\delta}/n \to 0$, for a fixed $r^2\mathcal{I}_r$.*

We prove Theorem 1.5 in Appendix D. The high-level technique we use is a standard one, showing that, it is statistically impossible to distinguish two slightly shifted copies of $f_r$ with probability $1 - \delta$, using $n$ samples. The shift corresponds to (twice) the estimation accuracy lower bound. The difficulty lies in getting the right constant.

Standard inequalities for showing indistinguishability results rely on calculating either the squared Hellinger distance [BY02] or the KL-divergence [BH79] between the two distributions. While these inequalities are straightforward to apply, given the calculated bounds on these statistical distances/divergences, the inequalities only yield constant-factor tightness in the estimation accuracy lower bound. On the other hand, in this work, we aim to give accuracy upper and lower bounds that are matching strongly, to within $1 + o(1)$ factors. As such, our proof of Theorem 1.5 involves delicate and non-standard bounding techniques which may be of independent interest. The proof techniques are currently slightly ad-hoc, and for future work, we hope to improve on these techniques to make them more general and more usable.

# 7 Experimental Results

In this section, we give experimental evidence supporting our proposed algorithmic theory. Our goals are to demonstrate that 1) $r$-smoothing is a beneficial pre-processing to the MLE, that $r$-smoothed Fisher information does capture the algorithmic performance in location estimation and 2) $r$-smoothed MLE can outperform the standard MLE, as well as standard mean estimation algorithms which do not leverage information about the distribution shape.

The version of smoothed MLE we use for experimentation is even simpler than Algorithm 1: use Gaussian smoothing before performing actual maximum likelihood finding over the entire real line, instead of returning a root of the empirical score function. This is closest to what statisticians would do in practice, and further does not require any initial uncertainty region on the true parameter $\lambda$.

We use the Gaussian-spiked *Laplace* model for experiments, with a Laplace distribution of density proportional to $e^{-|x|}$, and a Gaussian of mass 0.001 and width roughly 0.002 (the discretization granularity) added at $x = 4$. The reason we choose the Laplace over the Gaussian as the "body" of the distribution is because, fixing the variance of the distribution, the Laplace has twice the Fisher information as the Gaussian. Given that standard mean estimation algorithms only aim to achieve sub-Gaussian concentration, choosing the Laplace as the core distribution lets us demonstrate that the smoothed MLE can outperform mean estimation algorithms even in finite samples. We also note that this example is crucially different from the Dirac $\delta$-spiked example in Section 1.1. The Dirac $\delta$ spike has infinity density, whereas the narrow Gaussian spike only has somewhat large, but finite density. Given the finite and not too large density in the spike, in our experiments, the basic MLE algorithm will *not* "fit it at any cost", and instead has a smoother error distribution whenever we do not observe samples from the Gaussian spike. Nonetheless, even in this milder setting, we demonstrate that our smoothed MLE algorithm performs better than the original MLE.

Figure 2a is a heat map of the mean squared error of the smoothed MLE. The $x$-axis varies the number of samples $n$ from 50 to 5000, and the $y$-axis varies the smoothing parameter $r$ from 0.001 to 1 in log scale. Lighter color indicates a smaller mean squared error. The line overlaid on the heat map indicates, for each value of $n$, the value of $r$ with the smallest mean squared error. As $n$ increases, the optimal value of $r$ decreases, as predicted by our theory.

For small values of $n$—below about 1000—the mean squared error first decreases then increases again as we increase the smoothing parameter $r$. This confirms the theory in the paper: for small $n$, it is unlikely that we see any samples from the spike, in which case too small values for $r$ cause MLE to overfit. On the other hand, too large values of $r$ simply add too much noise, and also yields a sub-optimal mean squared error. The optimal value of $r$ is thus somewhere in between.

The situation changes when $n \gg 1000$, which is 1 over the mass of the spike. In this case, we expect to typically see samples from the spike, which allows us to estimate the mean highly accurately. Any smoothing just adds noise, and hence the optimal value of $r$ is close to 0.

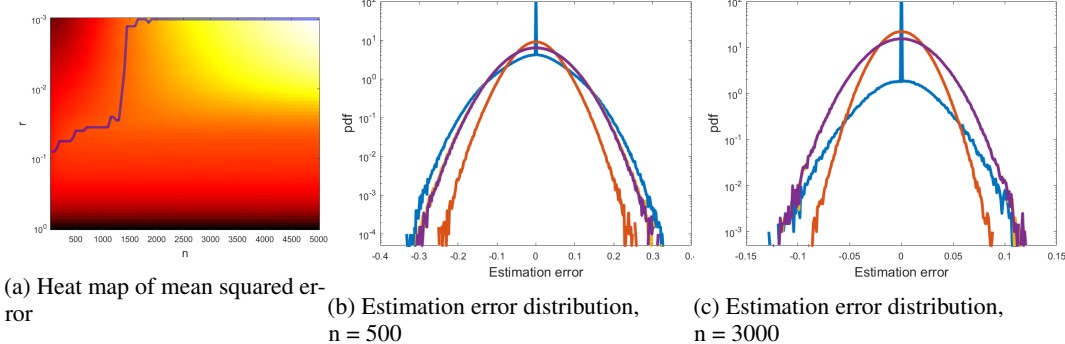

(a) Heat map of mean squared error

(b) Estimation error distribution, n = 500

(c) Estimation error distribution, n = 3000

Figure 2: Experimental results - Spiked Laplace model

Figure 2b picks $n = 500$ and compares the distribution of estimation errors across different algorithms: unsmoothed MLE (blue), 0.05-smoothed MLE (orange), empirical mean (yellow), Lee-Valiant (LV) estimator (purple) using $\delta = e^{-5}$. With only 500 samples, the unsmoothed MLE occasionally sees a sample from the spike, and attains high accuracy, but otherwise has large variance in error, compared with the empirical and LV estimators (which have essentially identical performance, so yellow is overlapped by purple on the plot). With just 0.05-smoothing, MLE outperforms all other estimators.

Figure 2c picks $n = 3000$ and compares the same algorithms. The unsmoothed MLE sees samples from the spike most of the time, and attains high accuracy, vastly outperforming the empirical and LV estimators (again, yellow is overlapped by purple). The 0.05-smoothed MLE performs worse than the unsmoothed MLE in the typical case, but has better tail behavior. This plot suggests that the optimal smoothing parameter $r$ in the high probability regime depends on the desired failure probability $\delta$.

# 8 Future Directions

One natural goal is to extend these techniques to estimate the mean of *unknown* distributions by means of a kernel density estimate (KDE), with accuracy dependent on the true distribution's Fisher information. In general this cannot work, a bias independent of the Fisher information is unavoidable, but for *symmetric* distributions the bias is zero and one can hope for good results. An asymptotic version of this was shown by Stone [Sto75], and we believe our techniques could get a finite-sample guarantee here.

A second direction is to investigate ways to generalize and simplify our lower bound analysis techniques. Recall that, while standard "indistinguishability" bounds based on squared Hellinger distance and KL-divergence are relatively straightforward to apply, they generally lose constant factors. Our analysis is tight to within a $1 + o(1)$-factor, but it requires analyzing several different parameters of the distribution. One can hope to extend and generalize these techniques to yield a new easy-to-apply bound, similar to those based on Hellinger distance and KL-divergence, that gives $1 + o(1)$-factor tightness.

# Acknowledgements

We thank the anonymous reviewers for insightful comments and suggestions on this work. Shivam Gupta and Eric Price are supported by NSF awards CCF-2008868, CCF-1751040 (CAREER), and the NSF AI Institute for Foundations of Machine Learning (IFML). Jasper C.H. Lee is supported in part by the generous funding of a Croucher Fellowship for Postdoctoral Research, NSF award DMS-2023239, NSF Medium Award CCF-2107079 and NSF AiTF Award CCF-2006206. Part of this work was done while Jasper C.H. Lee was a visitor at the Simons Institute for the Theory of Computing. Paul Valiant is supported by NSF award CCF-2127806.

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
