# A  Proof sketch of Proposition 1.1

In this appendix, we sketch the proof of Proposition 1.1.

**Proposition 1.1.** *The Gaussian+sawtooth model, with "teeth" of width $w$ and slope $\Delta$, has Fisher information $\Delta^2 \gg 1$ but no location estimator can have error $o(1/\sqrt{n})$ with constant probability over $n$ samples, unless $n > 0.01/w^2$.*

*This holds for arbitrarily large $\Delta$ and small $w$. By contrast, the asymptotic theory predicts error $O(1/(\Delta\sqrt{n}))$, which only holds for $n \gtrsim 1/w^2$.*

*Proof sketch.* Let $f$ denote the Gaussian with sawtooth model, and $f^{(\varepsilon)}$ denote the model but shifted by a distance of $\lceil \varepsilon/w \rceil w$. That is, the largest number of sawteeth that fits into a distance of $\varepsilon$.

Then, the KL divergence satisfies $D_{\mathrm{KL}}(f \parallel f^{(\varepsilon)}) \leq O(\varepsilon^2)$. By Pinsker's inequality, this implies that we need $\Omega(1/\varepsilon^2)$ samples to distinguish $f$ and $f^{(\varepsilon)}$ with constant probability. For $\varepsilon > w$, the shift between $f$ and $f^{(\varepsilon)}$ is at least $\varepsilon/2$.

Concluding, if $n < 0.01/w^2$, then we set $\varepsilon = 1/(10\sqrt{n})$ and there is no algorithm that can distinguish $f$ and $f^{(1/(10\sqrt{n}))}$ using $n$ samples with constant probability, meaning that no location estimator can achieve error $1/(20\sqrt{n})$ with constant probability. The reasoning generalizes to the high probability regime as well. $\qquad\square$

# B  Proofs from Section 3

We first prove a utility lemma, Lemma B.1, which we use throughout the rest of the paper.

**Lemma B.1.** *Let $f$ be an arbitrary distribution and let $f_r$ be the $r$-smoothed version of $f$. That is, $f_r = \mathbb{E}_{y \leftarrow f}\left[\frac{1}{\sqrt{2\pi r^2}} e^{-\frac{(x-y)^2}{2r^2}}\right]$. Let $s_r$ be the score function of $f_r$. Let $(X, Y, Z_r)$ be the joint distribution such that $Y \sim f$, $Z_r \sim \mathcal{N}(0, r^2)$ are independent, and $X = Y + Z_r \sim f_r$. We have, for every $\varepsilon > 0$,*

$$\frac{f_r(x+\varepsilon)}{f_r(x)} = \mathop{\mathbb{E}}_{Z_r \mid x}\left[e^{\frac{2\varepsilon Z_r - \varepsilon^2}{2r^2}}\right] \quad \text{and in particular} \quad s_r(x) = \mathop{\mathbb{E}}_{Z_r \mid x}\left[\frac{Z_r}{r^2}\right]$$

*Proof.* For simplicity of exposition, we only show the case where $f$ has a density. The general case can be proven by, for example, a limit argument. Let $w_r$ be the pdf of $\mathcal{N}(0, r^2)$. First, we show that for any $x, \varepsilon$ we have

$$\frac{f_r(x+\varepsilon)}{f_r(x)} = \mathop{\mathbb{E}}_{Z_r \mid x}\left[\frac{w_r(Z_r + \varepsilon)}{w_r(Z_r)}\right] \tag{2}$$

Denote the density of $(x, z, \hat{x})$ by $p(\cdot)$. Note that

$$p(z \mid x) = \frac{p(x, z)}{p(x)} = \frac{f(x-z)w_r(z)}{f_r(x)}$$

and hence

$$\begin{aligned}
f_r(x+\varepsilon) &= \int_{-\infty}^{\infty} w_r(z) f(x+\varepsilon-z)\,\mathrm{d}z = \int_{-\infty}^{\infty} w_r(z+\varepsilon) f(x-z)\,\mathrm{d}z \\
&= \int_{-\infty}^{\infty} p(z \mid x) f_r(x) \frac{w_r(z+\varepsilon)}{w_r(z)}\,\mathrm{d}z \\
&= f_r(x) \mathop{\mathbb{E}}_{Z \mid x}\left[\frac{w_r(Z_r + \varepsilon)}{w_r(Z_r)}\right]
\end{aligned}$$

proving (2).

Since $w_r(z) = \frac{1}{\sqrt{2\pi r^2}} e^{-\frac{z^2}{2r^2}}$, this gives

$$\frac{f_r(x+\varepsilon)}{f_r(x)} = \mathop{\mathbb{E}}_{Z \mid x}\left[e^{\frac{2\varepsilon Z_r - \varepsilon^2}{2r^2}}\right].$$

Taking the derivative with respect to $\varepsilon$ and evaluating at $\varepsilon = 0$,

$$\frac{f_r'(x)}{f_r(x)} = \mathop{\mathbb{E}}_{Z|x} \frac{Z_r}{r^2}.$$

$\square$

We now prove Lemmas 3.1 and 3.2, which upper and lower bound the $r$-smoothed Fisher information $\mathcal{I}_r$ respectively.

**Lemma 3.1.** *Let $\mathcal{I}_r$ be the Fisher information of an $r$-smoothed distribution $f_r$. Then, $\mathcal{I}_r \leq 1/r^2$.*

*Proof.* Using Lemma B.1 and Jensen's inequality,

$$I_r = \mathop{\mathbb{E}}_x[s_r^2(x)] = \mathop{\mathbb{E}}_x[(\mathop{\mathbb{E}}_{Z_r|x} Z_r/r^2)^2] \leq \mathop{\mathbb{E}}_x[\mathop{\mathbb{E}}_{Z_r|x} Z_r^2/r^4] = 1/r^2 \qquad \square$$

The lemma can alternatively be proven using Stam's inequality for Fisher information, which states that for independent real-valued random variables $X$ and $Y$, we have $1/\mathcal{I}(X + Y) \geq 1/\mathcal{I}(X) + 1/\mathcal{I}(Y)$.

**Lemma 3.2.** *Let $\mathcal{I}_r$ be the Fisher information for $f_r$, the $r$-smoothed version of distribution $f$. Let IQR be the interquartile range of $f$. Then, $\mathcal{I}_r \gtrsim 1/(\text{IQR} + r)^2$. Here, the hidden constant is a universal one independent of the distribution $f$ and independent of $r$.*

*Proof.* First, observe that $f_r$ is a smooth distribution in the sense that it is differentiable, and furthermore, its derivative is continuous. Thus, letting $R$ be the $30^{\text{th}}$-$70^{\text{th}}$ percentile range of $f_r$. Then, by a known result [SV11] (Section 3.1), $I_r \gtrsim 1/R^2$.

It then suffices to show that $R \leq \text{IQR} + O(r)$. Let $q_\ell$ be the $25^{\text{th}}$ percentile of $f$. Drawing a sample from $f_r$ is equivalent to independently drawing $x$ from $f$ and $z_r$ from $\mathcal{N}(0, r^2)$ and returning $x + z_r$. With probability at least 0.75, we have $x \geq q_\ell$. Also, with probability at least 0.95, we have $z_r \geq -\Theta(r)$. Therefore, by a union bound, we have $x + z_r \geq q_\ell - \Theta(r)$ except with probability at most 0.3, meaning that the $30^{\text{th}}$ percentile of $f_r$ is at least $q_\ell - \Theta(r)$. Combined with the symmetric argument for the $70^{\text{th}}$ percentile of $f_r$, this shows that $R \leq \text{IQR} + O(r)$. $\square$

Next, we prove another utility lemma, which states that the derivative of the score function cannot be too small for an $r$-smoothed distribution. Phrased differently, the score function of an $r$-smoothed distribution cannot decrease fast.

**Lemma B.2.** *$s_r'(x) \geq -1/r^2$ for all $x$, where $s_r$ is the score function of $f_r$, the $r$-smoothed version of distribution $f$.*

*Proof.* By taking the derivative of Lemma B.1 in $\varepsilon$,

$$\frac{f_r'(x + \varepsilon)}{f_r(x)} = \mathop{\mathbb{E}}_{Z|x}\left[e^{\frac{2\varepsilon Z - \varepsilon^2}{2r^2}} \frac{Z_r - \varepsilon}{r^2}\right]$$

Hence

$$s_r(x + \varepsilon) = \frac{f_r'(x + \varepsilon)}{f_r(x + \varepsilon)} = \frac{f_r'(x + \varepsilon)}{f_r(x)} \frac{f_r(x)}{f_r(x + \varepsilon)} = \frac{\mathbb{E}_{Z_r|x}\left[e^{\frac{2\varepsilon Z_r - \varepsilon^2}{2r^2}} \frac{Z_r - \varepsilon}{r^2}\right]}{\mathbb{E}_{Z_r|x}\left[e^{\frac{2\varepsilon Z_r - \varepsilon^2}{2r^2}}\right]}.$$

For $\varepsilon > 0$, since $e^{\frac{2\varepsilon Z_r - \varepsilon^2}{2r^2}}$ and $\frac{Z_r - \varepsilon}{r^2}$ are monotonically increasing in $Z_r$, and the former is nonnegative, they are positively correlated:

$$\mathop{\mathbb{E}}_{Z_r|x}\left[e^{\frac{2\varepsilon Z_r - \varepsilon^2}{2r^2}} \frac{Z_r - \varepsilon}{r^2}\right] \geq \mathop{\mathbb{E}}_{Z_r|x}\left[e^{\frac{2\varepsilon Z_r - \varepsilon^2}{2r^2}}\right] \mathop{\mathbb{E}}_{Z_r|x}\left[\frac{Z_r - \varepsilon}{r^2}\right]$$

Hence

$$s_r(x + \varepsilon) \geq \mathop{\mathbb{E}}_{Z_r|x}\left[\frac{Z_r - \varepsilon}{r^2}\right] = s_r(x) - \frac{\varepsilon}{r^2}.$$

or (taking $\varepsilon \to 0$), $s_r'(x) \geq -\frac{1}{r^2}$. $\square$

Lastly, we prove the concentration of empirical score function. The way we do so is to show (Lemma B.6) that the $k^{\text{th}}$ absolute moment of the score function is upper bounded according to the standard moment bounds for sub-Gamma distributions. As a corollary (Corollary 3.3), we get that the scores have sub-Gamma concentration.

As a utility lemma, we bound the moments of the score function when the score function is aligned with the distribution, instead of being misaligned by some $\varepsilon$ distance.

**Lemma B.3.** *Let $s_r$ be the score function of an $r$-smoothed distribution $f_r$ with Fisher information $\mathcal{I}_r$. Then, for $k \geq 3$,*

$$\mathbb{E}_x[|s_r(x)|^k] \leq (1.6/r)^{k-2}k^{k/2}\mathcal{I}_r$$

*Proof.* For any $x, \varepsilon$, by Lemma B.1 and Jensen's inequality,

$$f_r(x+\varepsilon) \geq f_r(x)e^{\varepsilon s_r(x)-\frac{\varepsilon^2}{2r^2}}.$$

Setting $\varepsilon = \pm r$ with sign matching $s_r(x)$, we have that

$$f_r(x + r\text{sign}(s_r(x))) \geq f_r(x)e^{r|s_r(x)|}/\sqrt{e}.$$

We also have, from Lemma B.2, that

$$s_r(x-r) \leq s_r(x) + 1/r$$

and

$$s_r(x+r) \geq s_r(x) - 1/r.$$

In other words,

$$|s_r(x+r\text{sign}(s_r(x)))| \geq |s_r(x)| - 1/r.$$

Therefore, for any $k \geq 2$, and $|s_r(x)| > \alpha/r$ for $\alpha := 2 + 1.2\sqrt{k}$,

$$
\begin{aligned}
f_r(x+r\text{sign}(s_r(x)))|s_r(x+r\text{sign}(s_r(x)))|^k &\geq \frac{1}{\sqrt{e}}f_r(x)e^{r|s_r(x)|}(|s_r(x)|-1/r)^k \\
&= f_r(x)|s_r(x)|^k \cdot \left(\frac{1}{\sqrt{e}}e^{r|s_r(x)|}(1-\frac{1}{r|s_r(x)|})^k\right) \\
&\geq f_r(x)|s_r(x)|^k \cdot \left(\frac{1}{\sqrt{e}}e^{\alpha-1.4\frac{k}{\alpha}}\right) \\
&\geq f_r(x)|s_r(x)|^k \cdot 4.
\end{aligned}
$$

Therefore

$$f_r(x)|s_r(x)|^k \leq \frac{1}{4}\left(f_r(x-r)|s_r(x-r)|^k + f_r(x+r)|s_r(x+r)|^k\right) \tag{3}$$

whenever $k \geq 2$ and $|s_r(x)| \geq \alpha/r$. Integrating this,

$$
\begin{aligned}
\mathbb{E}[s_r^k(x)] = \int_{-\infty}^{\infty} f_r(x)|s_r(x)|^k \, \mathrm{d}x &= 2\int_{-\infty}^{\infty} f_r(x)|s_r(x)|^k - \frac{1}{4}f_r(x-r)|s_r(x-r)|^k - \frac{1}{4}f_r(x+r)|s_r(x+r)|^k \, \mathrm{d}x \\
&\leq 2\int_{-\infty}^{\infty} f_r(x)|s_r(x)|^k 1_{|s_r(x)|<\alpha/r} \, \mathrm{d}x \\
&\leq 2\int_{-\infty}^{\infty} f_r(x)|s_r(x)|^2(\alpha/r)^{k-2}1_{|s_r(x)|<\alpha/r} \, \mathrm{d}x \\
&\leq 2(\alpha/r)^{k-2} \mathbb{E}_x[s_r^2(x)] = 2(\alpha/r)^{k-2}\mathcal{I}_r
\end{aligned}
$$

Finally, we observe for any $k \geq 2$ that

$$2(1.2\sqrt{k}+2)^{k-2} \leq k^{k/2} \cdot 1.6^{k-2}$$

giving the lemma.

$\square$

The proof of Lemma B.6 has the same logical structure as the proof of Lemma B.3, but has further subtleties. The following two lemmas generalize the first step in the proof of Lemma B.3.

**Lemma B.4.** *Let $s_r$ be the score function of an $r$-smoothed distribution $f_r$ with Fisher information $\mathcal{I}_r$. For any $x$, $k \geq 3$ and $0 \leq \varepsilon \leq r/2$, if $s_r(x+\varepsilon) \geq \max(2\sqrt{k}+2, 9.5)/r$, then*

$$f_r(x)|s_r(x+\varepsilon)|^k \leq \frac{1}{5}\max\left(f_r(x-\varepsilon)|s_r(x-\varepsilon)|^k, f_r(x+\varepsilon+r)|s_r(x+\varepsilon+r)|^k\right)$$

*Proof.* Let $\alpha := \frac{f_r(x)}{f_r(x+\varepsilon)}$. By Lemma B.1, we have

$$\alpha = \mathop{\mathbb{E}}_{Z_r|x+\varepsilon}\left[e^{\frac{-2\varepsilon Z_r - \varepsilon^2}{2r^2}}\right] \tag{4}$$

We will consider two cases.

**When $\log\alpha \leq \frac{3}{4}rs_r(x+\varepsilon)-2$.** First, by Lemma B.1 and Jensen's inequality, we have

$$\frac{f_r(x+\varepsilon+r)}{f_r(x+\varepsilon)} \geq e^{rs_r(x+\varepsilon)-1/2}$$

We also have, by Lemma B.2,

$$s_r(x+\varepsilon+r) \geq s_r(x+\varepsilon)-1/r$$

So,

$$f_r(x+\varepsilon+r)|s_r(x+\varepsilon+r)|^k \geq f_r(x+\varepsilon)|s_r(x+\varepsilon)|^k e^{rs_r(x)-1/2}\left(1-\frac{1}{rs_r(x+\varepsilon)}\right)^k$$

$$\geq f_r(x+\varepsilon)|s_r(x+\varepsilon)|^k e^{rs_r(x+\varepsilon)-\frac{k}{rs_r(x+\varepsilon)-1}-1/2}$$

Since $s_r(x+\varepsilon) \geq (2\sqrt{k}+2)/r$,

$$f_r(x+\varepsilon+r)|s_r(x+\varepsilon+r)|^k \geq f_r(x+\varepsilon)|s_r(x+\varepsilon)|^k e^{\frac{3}{4}rs_r(x+\varepsilon)}$$

So, since

$$\alpha = \frac{f_r(x)}{f_r(x+\varepsilon)} \leq e^{\frac{3}{4}rs_r(x+\varepsilon)-2}$$

we have

$$f(x)|s_r(x+\varepsilon)|^k = \alpha f_r(x+\varepsilon)|s_r(x+\varepsilon)|^k \leq \frac{1}{5}f(x+\varepsilon+r)|s_r(x+\varepsilon+r)|^k$$

**When $\log\alpha > \frac{3}{4}rs_r(x+\varepsilon)-2$** Evaluating (4) at $x-\varepsilon$ gives

$$\frac{f_r(x-\varepsilon)}{f_r(x)} = \mathop{\mathbb{E}}_{Z_r|x}\left[e^{\frac{-2\varepsilon Z_r - \varepsilon^2}{2r^2}}\right]$$

Taking derivative with respect to $\varepsilon$, we have

$$\frac{f_r'(x-\varepsilon)}{f_r(x)} = \mathop{\mathbb{E}}_{Z_r|x}\left[\frac{(Z_r+\varepsilon)}{r^2}e^{\frac{-2\varepsilon Z_r - \varepsilon^2}{2r^2}}\right]$$

and so by evaluating at $x+\varepsilon$ (to "shift back")

$$\frac{f_r'(x)}{f_r(x+\varepsilon)} = \mathop{\mathbb{E}}_{Z_r|x+\varepsilon}\left[\frac{(Z_r+\varepsilon)}{r^2}e^{\frac{-2\varepsilon Z_r - \varepsilon^2}{2r^2}}\right]$$

Define $y = e^{\frac{-2\varepsilon Z_r - 2\varepsilon^2}{2r^2}}$, so that $\mathbb{E}_{Z_r|x+\varepsilon}[y] = \alpha e^{\frac{\varepsilon^2}{2r^2}}$, and

$$\frac{(Z_r+\varepsilon)}{r^2}e^{\frac{-2\varepsilon Z_r - \varepsilon^2}{2r^2}} = -\frac{e^{\frac{\varepsilon^2}{2r^2}}}{\varepsilon}y\log y$$

is concave, so by Jensen's inequality

$$\frac{f'(x)}{f(x+\varepsilon)} \le -\frac{e^{\varepsilon^2/(2r^2)}}{\varepsilon}(e^{-\frac{\varepsilon^2}{2r^2}}\alpha)\log(e^{-\frac{\varepsilon^2}{2r^2}}\alpha) = -\frac{\alpha\log\alpha}{\varepsilon} + \frac{\alpha\varepsilon}{2r^2}$$

So,

$$s_r(x) = \frac{f'_r(x)}{f_r(x)} \le -\frac{\log\alpha}{\varepsilon} + \frac{\varepsilon}{2r^2}$$

Finally, we move to consider the point $x - \varepsilon$. By Lemma B.2, we have

$$s_r(x-\varepsilon) \le s_r(x) + \varepsilon/r^2 \le -\frac{\log\alpha}{\varepsilon} + \frac{3\varepsilon}{2r^2}$$

By Lemma B.1,

$$\frac{f(x-\varepsilon)}{f(x+\varepsilon)} = \mathop{\mathbb{E}}_{Z_r|x+\varepsilon}\left[e^{-\frac{4\varepsilon Z_r - 4\varepsilon^2}{2r^2}}\right] = \mathop{\mathbb{E}}_{Z_r|x+\varepsilon}[y^2] \ge \mathop{\mathbb{E}}_{Z_r|x+\varepsilon}[y]^2 = \alpha^2 e^{-\frac{\varepsilon^2}{r^2}}$$

Since $\log\alpha \ge \frac{3}{4}s_r(x+\varepsilon) - 2$,

$$-s_r(x-\varepsilon) \ge \frac{\frac{3}{4}rs_r(x+\varepsilon) - 2}{\varepsilon} - \frac{3\varepsilon}{2r^2} \ge \frac{3}{2}s_r(x+\varepsilon) - \frac{4.75}{r} \ge s_r(x)$$

where the second inequality comes from the fact that $\frac{3}{4}rs_r(x+\varepsilon) - 2 > 0$ and so the function is decreasing in $\varepsilon$, with minimum evaluated at $\varepsilon = r/2$.

Thus, we have

$$f_r(x-\varepsilon)|s_r(x-\varepsilon)|^k \ge \alpha e^{-\varepsilon^2/r^2} f_r(x)|s_r(x+\varepsilon)|^k$$

Since our assumptions give $\alpha e^{-\varepsilon^2/r^2} \ge e^{5.125}e^{-1/4} \ge 5$, we get the result. $\qquad\square$

**Lemma B.5.** *Let $s_r$ be the score function of an $r$-smoothed distribution $f_r$ with Fisher information $\mathcal{I}_r$.*

*For any $x$, $k \ge 3$ and $-r/2 \le \varepsilon \le 0$, if $s_r(x+\varepsilon) \ge \alpha/r$ for $\alpha = 2 + 1.2\sqrt{k}$, then we have*

$$f_r(x)|s_r(x+\varepsilon)|^k \le \frac{1}{4}\left(f_r(x-r)|s_r(x+\varepsilon-r)|^k + f_r(x+r)|s_r(x+\varepsilon+r)|^k\right)$$

*As an immediate corollary, the statement is true also when $\varepsilon \in [0, r/2)$ and $s_r(x) \le -\alpha/r$.*

*Proof.* For any $x, \kappa$, by Lemma B.1 and Jensen's inequality,

$$f_r(x+\kappa) \ge f_r(x)e^{\kappa s_r(x) - \frac{\kappa^2}{2r^2}}.$$

So, setting $\kappa = r$, we have

$$f_r(x+r) \ge f_r(x)e^{rs_r(x)}/\sqrt{e}$$

By Lemma B.2, we have that

$$s_r(x+\varepsilon+r) \ge s_r(x+\varepsilon) - 1/r$$

Since our right hand side is positive by assumption, this is equivalently stated as

$$|s_r(x+\varepsilon+r))| \ge |s_r(x+\varepsilon)| - 1/r$$

When $\varepsilon < 0$, we have, by Lemma B.2, and since $|\varepsilon| \le r$, $s_r(x) \ge s_r(x+\varepsilon) - 1/r$. So,

$$\begin{aligned}
f_r(x+r)|s_r(x+\varepsilon+r)|^k &\ge \frac{1}{\sqrt{e}}f_r(x)e^{rs_r(x)}(|s_r(x+\varepsilon)| - 1/r)^k \\
&\ge \frac{1}{\sqrt{e}}f_r(x)e^{r(s_r(x+\varepsilon)-1/r)}(|s_r(x+\varepsilon)| - 1/r)^k \\
&\ge f_r(x)|s_r(x+\varepsilon)|^k\left(\frac{1}{\sqrt{e}}e^{r|s_r(x+\varepsilon)|-1}\left(1 - \frac{1}{r|s_r(x+\varepsilon)|}\right)^k\right) \\
&\ge f_r(x)|s_r(x+\varepsilon)|^k \cdot \left(e^{-3/2}e^{\alpha - 1.4k/\alpha}\right) \\
&\ge f_r(x)|s_r(x+\varepsilon)|^k \cdot 4 \quad \text{since } k \ge 3
\end{aligned}$$

$\qquad\square$

We are now ready to prove Lemma B.6, which states that the distribution of the score function $s_r(x + \varepsilon)$ where $x \sim f_r$ is a sub-Gamma distribution. Corollary 3.3 then states that the average of many score function samples is well-concentrated, following sub-Gamma concentration.

**Lemma B.6.** *Let $s_r$ be the score function of an $r$-smoothed distribution $f_r$ with Fisher information $\mathcal{I}_r$. Then, for $k \geq 3$ and $|\varepsilon| \leq r/2$,*

$$\mathbb{E}_x[|s_r(x+\varepsilon)|^k] \leq \frac{k!}{2}(15/r)^{k-2}\max(\mathbb{E}_x[s_r^2(x+\varepsilon)], \mathcal{I}_r)$$

*Equivalently, $s_r(x + \varepsilon)$ is a sub-Gamma random variable.*

$$s_r(x+\varepsilon) \in \Gamma(\max_x(\mathbb{E}[s_r^2(x+\varepsilon)], \mathcal{I}_r), 15/r).$$

*Proof.* Without loss of generality we only show the $\varepsilon \geq 0$ case.

Using Lemma B.4 and Lemma B.3, we have

$$\int_{-\infty}^{\infty} f_r(x-\varepsilon)|s_r(x)|^k \mathbb{1}_{s_r(x)>\max(2\sqrt{k}+2,9.5)/r} \, dx$$

$$\leq \frac{1}{5}\int_{-\infty}^{\infty} f_r(x-2\varepsilon)|s_r(x-2\varepsilon)|^k + f_r(x+r)|s_r(x+r)|^k \, dx$$

$$= \frac{2}{5}\mathbb{E}[|s_r(x)|^k]$$

$$\leq \frac{2}{5}(1.6/r)^{k-2}k^{k/2}\mathcal{I}_r$$

We can start bounding the $k^{\text{th}}$ moment quantity in the lemma:

$$\mathbb{E}[|s_r(x+\varepsilon)|^k]$$

$$= \int_{-\infty}^{\infty} f_r(x-\varepsilon)|s_r(x)|^k \, dx$$

$$= 2\int_{-\infty}^{\infty} f_r(x-\varepsilon)|s_r(x)|^k - \frac{1}{4}f_r(x-\varepsilon-r)|s_r(x-r)|^k - \frac{1}{4}f_r(x-\varepsilon+r)|s_r(x+r)|^k \, dx$$

$$\leq 2\int_{-\infty}^{\infty} f_r(x-\varepsilon)|s_r(x)|^k \mathbb{1}_{s_r(x)\geq -\max(2\sqrt{k}+2,9.5)/r} \, dx$$

where the last inequality follows from (a slight weakening of) Lemma B.5. Now, by the previous claim, we get that

$$\mathbb{E}[|s_r(x+\varepsilon)|^k]$$

$$\leq 2\int_{-\infty}^{\infty} f_r(x-\varepsilon)|s_r(x)|^k \mathbb{1}_{|s_r(x)|\leq\max(2\sqrt{k}+2,9.5)/r} \, dx + \frac{4}{5}(1.6/r)^{k-2}k^{k/2}\mathcal{I}_r$$

$$\leq 2\int_{-\infty}^{\infty} f_r(x-\varepsilon)|s_r(x)|^2(\max(2\sqrt{k}+2,9.5)/r)^{k-2}\mathbb{1}_{|s_r|\leq\max(2\sqrt{k}+2,9.5)/r} \, dx + \frac{4}{5}(1.6/r)^{k-2}k^{k/2}\mathcal{I}_r$$

$$\leq 2(\max(2\sqrt{k}+2,9.5)/r)^{k-2}\mathbb{E}_x[|s_r(x+\varepsilon)|^2] + \frac{4}{5}(1.6/r)^{k-2}k^{k/2}\mathcal{I}_r$$

$$\leq 2k^{k/2}(2.5/r)^{k-2}\mathbb{E}_x[|s_r(x+\varepsilon)|^2] + \frac{4}{5}(1.6/r)^{k-2}k^{k/2}\mathcal{I}_r$$

$$\leq 3k^{k/2}(2.5/r)^{k-2}\max(\mathbb{E}_x[|s_r(x+\varepsilon)|^2], \mathcal{I}_r)$$

$$\leq \frac{k!}{2}(15/r)^{k-2}\max(\mathbb{E}_x[|s_r(x+\varepsilon)|^2], \mathcal{I}_r)$$

$\square$

**Corollary 3.3.** *Let $f$ be an arbitrary distribution and let $f_r$ be the $r$-smoothed version of $f$. That is,*
$$f_r(x) = \mathbb{E}_{y\leftarrow f}\left[\frac{1}{\sqrt{2\pi r^2}}e^{-\frac{(x-y)^2}{2r^2}}\right]. \text{ Consider the parametric family of distributions } f_r^\lambda(x) = f_r(x-\lambda).$$

*Suppose we take $n$ i.i.d. samples $y_1, \ldots, y_n \leftarrow f_r^\lambda$, and consider the empirical score function $\hat{s}$ mapping a candidate parameter $\hat{\lambda}$ to $\frac{1}{n}\sum_i s_r(y_i - \hat{\lambda})$, where $s_r$ is the score function of $f_r$.*

*Then, for any $|\varepsilon| \leq r/2$,*

$$\Pr_{y_i \overset{i.i.d.}{\sim} f_r^\lambda} \left( |\hat{s}(\lambda + \varepsilon) - \mathbb{E}_{x \leftarrow f_r}[s(x - \varepsilon)]| \geq \sqrt{\frac{2\max(\mathbb{E}_x[s_r^2(x-\varepsilon)], \mathcal{I}_r)\log\frac{2}{\delta}}{n}} + \frac{15\log\frac{2}{\delta}}{nr} \right) \leq \delta$$

*Proof.* Since $\hat{s}(\lambda + \varepsilon) = \frac{1}{n}\sum_{i=1}^n s_r(y_i - \lambda - \varepsilon) = \frac{1}{n}\sum_{i=1}^n s_r(x_i - \varepsilon)$, we know that by Lemma B.6 and the standard algebra of sub-Gamma distributions that $\hat{s}(\lambda + \varepsilon) \in \Gamma(\frac{1}{n}\max(\mathbb{E}_x[s_r^2(x + \varepsilon)], \mathcal{I}_r), 15/r)$. The corollary then follows from the standard Bernstein inequality for sub-Gamma distributions [BLM13]. $\qquad\square$

## C  Proofs omitted in Section 4

We first give the proof of Theorem 1.3, assuming Theorem 4.1.

*Proof of Theorem 1.3.* It suffices to show that conditions 2) and 3) in the corollary statement implies that each of the following terms from Theorem 4.1 is $1 + O(1/\gamma)$:

- $1 + 1/\log\frac{1}{\delta}$: Note that $\mathcal{I}_r \leq \frac{1}{r^2}$ by Lemma 3.1 and so condition 3) implies that $1/\log\frac{1}{\delta} \leq (\log\log\frac{1}{\delta})/\log\frac{1}{\delta} \leq \frac{1}{\gamma}$

- $1 + \rho_r$: $\sqrt{1 + O(1/\gamma)} = 1 + O(1/\gamma)$. It suffices to check that $\left(\frac{2\log\frac{4\log\frac{1}{\delta}}{r^2\mathcal{I}_r(1-\frac{\beta}{\gamma})\delta}}{n}\right)^{\frac{1}{4}} = O(1/\sqrt{\gamma})$. The fact that $\log\frac{\log\frac{1}{\delta}}{\delta} = O(\log\frac{1}{\delta})$ together with condition 3) imply that the quantity is bounded by $O\left(\left(\frac{(1+O(\gamma))\log\frac{1}{\delta}}{n}\right)^{\frac{1}{4}}\right)$, which in turn is bounded by $O(1/\sqrt{\gamma})$ by condition 2).

- $1/(1 - \beta/\gamma) \leq 1 + O(1/\gamma)$ since $\beta$ is a constant

- $\sqrt{1 + \frac{\log\frac{4\log\frac{1}{\delta}}{r^2\mathcal{I}_r(1-\frac{\beta}{\gamma})}}{\log\frac{1}{\delta}}}$: Note that $\frac{\log\frac{4\log\frac{1}{\delta}}{1-\frac{\beta}{\gamma}}}{\log\frac{1}{\delta}} = O(\frac{\log\log\frac{1}{\delta}}{\log\frac{1}{\delta}}) = O(1/\gamma)$ as before. Also, condition 3) implies that $(\log\frac{1}{r^2\mathcal{I}_r})/(\log\frac{1}{\delta}) \leq (\log\frac{1}{r^2\mathcal{I}_r})(\log\log\frac{1}{\delta})/(\log\frac{1}{\delta}) \leq 1/\gamma$.

$\qquad\square$

The rest of this appendix is on proving Lemma 4.2, which via reparameterization gives Theorem 4.1.

We first show a utility lemma (Lemma C.1), before using it to prove Lemmas C.2 and C.3, which bound the expectation and variance of the empirical score function. After that, we prove Lemma 4.2.

**Lemma C.1.** *Let $w_r$ be a Gaussian with standard deviation $r$, $f$ be an arbitrary probability distribution, and $f_r$ be the $r$-smoothed version of $f$. Define*

$$\Delta_\varepsilon(x) := \frac{f_r(x+\varepsilon) - f_r(x) - \varepsilon f_r'(x)}{f_r(x)}.$$

*Then for any $|\varepsilon| \leq r/2$,*

$$\mathbb{E}_{x\sim f_r}\left[\Delta_\varepsilon(x)^2\right] \lesssim \frac{\varepsilon^4}{r^4}.$$

*Proof.* By Lemma B.1, we have

$$\Delta_\varepsilon(x) = \frac{f_r(x+\varepsilon) - f_r(x) - \varepsilon f_r'(x)}{f_r(x)} = \mathop{\mathbb{E}}_{Z_r|x} \left( e^{\frac{2\varepsilon Z_r - \varepsilon^2}{2r^2}} - 1 - \frac{\varepsilon Z_r}{r^2} \right).$$

Define

$$\alpha_\varepsilon(z) := e^{\frac{2\varepsilon z - \varepsilon^2}{2r^2}} - 1 - \frac{\varepsilon z}{r^2}.$$

We want to bound

$$\begin{aligned}
&\mathop{\mathbb{E}}_X \left[ \Delta_\varepsilon(x)^2 \right] \\
&= \mathop{\mathbb{E}}_X \left[ \mathop{\mathbb{E}}_{Z_r|X} \left[ \alpha_\varepsilon(Z_r) \right]^2 \right] \\
&\le \mathop{\mathbb{E}}_{X,Z_r} \left[ (\alpha_\varepsilon(Z_r))^2 \right] \\
&= \mathop{\mathbb{E}}_{Z_r \sim N(0,r^2)} (\alpha_\varepsilon(Z_r))^2.
\end{aligned} \tag{5}$$

Finally, we bound this term (5).

When $|\varepsilon z| \le r^2$, we have by a Taylor expansion that

$$e^{\frac{2\varepsilon z - \varepsilon^2}{2r^2}} = 1 + \frac{\varepsilon z}{r^2} - \frac{\varepsilon^2}{2r^2} + O\left( \left( \frac{2\varepsilon z - \varepsilon^2}{2r^2} \right)^2 \right)$$

and so

$$|\alpha_\varepsilon(z)| \lesssim \frac{\varepsilon^2}{r^2} + \left( \frac{\varepsilon z}{r^2} \right)^2$$

This implies that $(\alpha_\varepsilon(z))^2 \lesssim \varepsilon^4/r^4 + \varepsilon^4 z^4/r^8$, meaning that

$$\begin{aligned}
\mathop{\mathbb{E}}_{Z_r \sim N(0,r^2)} \left( \alpha_\varepsilon(Z_r)^2 \cdot \mathbb{1}_{|\varepsilon Z_r| \le r^2} \right) &\lesssim \mathop{\mathbb{E}}_{Z_r \sim N(0,r^2)} \left( \left( \frac{\varepsilon^4}{r^4} + \frac{\varepsilon^4 z^4}{r^8} \right) \cdot \mathbb{1}_{|\varepsilon Z_r| \le r^2} \right) \\
&\lesssim \frac{\varepsilon^4}{r^4} + \mathop{\mathbb{E}}_{Z_r \sim N(0,r^2)} \left( \frac{\varepsilon^4 z^4}{r^8} \right) \\
&\lesssim \frac{\varepsilon^4}{r^4}.
\end{aligned} \tag{6}$$

On the other hand, in the case where $|\varepsilon z| \ge r^2$, we have the following inequality:

$$|\alpha_\varepsilon(z)| \le e^{|\frac{\varepsilon z}{r^2}|}$$

so

$$\begin{aligned}
\mathop{\mathbb{E}}_{Z_r \sim N(0,r^2)} \left( \alpha_\varepsilon(Z_r)^2 \cdot \mathbb{1}_{|\varepsilon Z_r| \ge r^2} \right) &\le 2 \int_{|r^2/\varepsilon|}^\infty \frac{1}{\sqrt{2\pi r^2}} e^{\frac{2|\varepsilon z|}{r^2}} e^{-\frac{z^2}{2r^2}} \, \mathrm{d}z \\
&= 2 e^{2\varepsilon^2/r^2} \int_{|r^2/\varepsilon|}^\infty \frac{1}{\sqrt{2\pi r^2}} e^{-\frac{(z-2|\varepsilon|)^2}{2r^2}} \, \mathrm{d}z \\
&\le 2\sqrt{e} \Pr[z \ge r^2/|\varepsilon| - 2|\varepsilon|] \\
&\lesssim e^{-\frac{(|r^2/\varepsilon| - 2|\varepsilon|)^2}{2r^2}} \\
&\le e^{-\frac{r^2}{8\varepsilon^2}} \lesssim \frac{\varepsilon^4}{r^4}.
\end{aligned}$$

Which combines with (5) and (6) to give the result. $\qquad \square$

We are now ready to prove Lemma C.2, which bounds the expectation of the empirical score function.

**Lemma C.2.** *Suppose $f_r$ is an $r$-smoothed distribution with Fisher information $\mathcal{I}_r$. Then, for any $|\varepsilon| \le r/2$, the expected score $\mathbb{E}_{x \sim f_r}[s_r(x + \varepsilon)]$ satisfies*

$$\mathop{\mathbb{E}}_{x \sim f_r}[s_r(x + \varepsilon)] = -\mathcal{I}\varepsilon + \Theta\left(\sqrt{\mathcal{I}_r}\frac{\varepsilon^2}{r^2}\right)$$

*Proof.* By definition of $s_r$,

$$\mathop{\mathbb{E}}_{x \sim f_r}[s_r(x + \varepsilon)] = \int_{-\infty}^{\infty} \frac{f_r(x) f_r'(x + \varepsilon)}{f_r(x + \varepsilon)}\, \mathrm{d}x$$

$$= \int_{-\infty}^{\infty} f_r'(x) \frac{f_r(x - \varepsilon) - f_r(x)}{f_r(x)}\, \mathrm{d}x$$

Since by definition of $\mathcal{I}_r$,

$$\mathcal{I}_r := \int_{-\infty}^{\infty} \frac{f_r'(x)^2}{f_r(x)}\, \mathrm{d}x,$$

$$\mathop{\mathbb{E}}_{x \sim f_r}[s_r(x + \varepsilon)] + \varepsilon \mathcal{I}_r = \int_{-\infty}^{\infty} \frac{f_r'(x)}{f_r(x)}(f_r(x - \varepsilon) - f_r(x) + \varepsilon f_r'(x))\, \mathrm{d}x$$

$$= \mathbb{E}[s_r(x) \cdot \Delta_{-\varepsilon}(x)]$$

where $\Delta_\varepsilon(x) := \frac{f_r(x + \varepsilon) - f_r(x) - \varepsilon f_r'(x)}{f_r(x)}$. Thus

$$\left(\mathop{\mathbb{E}}_{x \sim f_r}[s_r(x + \varepsilon)] + \varepsilon \mathcal{I}_r\right)^2 \le \mathbb{E}\left[s_r(x)^2\right] \mathbb{E}\left[\Delta_{-\varepsilon}(x)^2\right]$$

$$= \mathcal{I}_r \mathbb{E}\left[\Delta_{-\varepsilon}(x)^2\right]$$

By Lemma C.1, we have that

$$\mathbb{E}\left[\Delta_\varepsilon(x)^2\right] \lesssim \frac{\varepsilon^4}{r^4}.$$

and so

$$\left|\mathop{\mathbb{E}}_{x \sim f_r}[s_r(x + \varepsilon)] + \varepsilon \mathcal{I}_r\right| \lesssim \sqrt{\mathcal{I}_r}\frac{\varepsilon^2}{r^2}$$

as desired. $\qquad\square$

**Lemma C.3.** *Suppose $f_r$ is an $r$-smoothed distribution with Fisher information $\mathcal{I}_r$. For any $|\varepsilon| \le r/2$, if $r/\varepsilon \gtrsim \sqrt{\log e/(r^2\mathcal{I}_r)}$, then the second moment of the score satisfies*

$$\mathop{\mathbb{E}}_{x \sim f_r}[s_r^2(x + \varepsilon)] \le \mathcal{I}_r + O\left(\frac{\varepsilon}{r}\mathcal{I}_r\sqrt{\log\frac{e}{r^2\mathcal{I}_r}}\right)$$

*Proof.* We have that

$$\mathop{\mathbb{E}}_{x \sim f_r}[s_r^2(x + \varepsilon)] = \int_{-\infty}^{\infty} f_r(x)\left(\frac{f_r'(x + \varepsilon)}{f_r(x + \varepsilon)}\right)^2\, \mathrm{d}x$$

$$= \int_{-\infty}^{\infty} f_r(x - \varepsilon)\left(\frac{f_r'(x)}{f_r(x)}\right)^2\, \mathrm{d}x$$

$$= \mathcal{I}_r + \int_{-\infty}^{\infty} (f_r(x - \varepsilon) - f_r(x))\left(\frac{f_r'(x)}{f_r(x)}\right)^2\, \mathrm{d}x$$

By Lemma B.1, we have

$$\frac{f_r(x - \varepsilon) - f_r(x)}{f_r(x)} = \mathop{\mathbb{E}}_{Z_r | x}\left(e^{-\varepsilon Z_r/r^2 - \varepsilon^2/2r^2} - 1\right).$$

We have that

$$\frac{f_r'(x)}{f_r(x)} = \mathop{\mathbb{E}}_{Z_r | x}\frac{Z_r}{r^2}.$$

so that we need to bound

$$\mathop{\mathbb{E}}_{x \sim f_r}\left[s_r^2(x+\varepsilon)\right] - \mathcal{I}_r = \mathop{\mathbb{E}}_{x}\left[\left(\mathop{\mathbb{E}}_{Z_r|x}(e^{-\varepsilon Z_r/r^2 - \varepsilon^2/2r^2} - 1)\right)\left(\mathop{\mathbb{E}}_{Z_r|x}\frac{Z_r}{r^2}\right)^2\right]. \qquad (7)$$

We can bound this as follows. For any $x$ and parameter $\alpha \lesssim r/\varepsilon$,

$$\left(\mathop{\mathbb{E}}_{Z_r|x}(e^{-\varepsilon Z_r/r^2 - \varepsilon^2/2r^2} - 1)\right)\left(\mathop{\mathbb{E}}_{Z_r|x}\frac{Z_r}{r^2}\right)^2 \le \left(O(\varepsilon\alpha/r) + \mathop{\mathbb{E}}_{Z_r|x}\mathbb{1}_{|Z_r|>\alpha r}(e^{-\varepsilon Z_r/r^2} - 1)\right)\left(\mathop{\mathbb{E}}_{Z_r|x}\frac{Z_r}{r^2}\right)^2$$

$$\le O(\varepsilon\alpha/r)s_r^2(x) + \left(\mathop{\mathbb{E}}_{Z_r|x}\mathbb{1}_{|Z_r|>\alpha r}(e^{-\varepsilon Z_r/r^2} - 1)\right)\left(\mathop{\mathbb{E}}_{Z_r|x}\left[\frac{Z_r^2}{r^4}\right]\right)$$

Thus

$$\mathop{\mathbb{E}}_{x \sim f_r}\left[s_r^2(x+\varepsilon)\right] - \mathcal{I}_r \le O(\varepsilon\alpha/r)\mathcal{I}_r + \mathop{\mathbb{E}}_{x}\left[\left(\mathop{\mathbb{E}}_{Z_r|x}\mathbb{1}_{|Z_r|>\alpha r}(e^{-\varepsilon Z_r/r^2} - 1)\right)\left(\mathop{\mathbb{E}}_{Z_r|x}\left[\frac{Z_r^2}{r^4}\right]\right)\right]$$

It remains to bound the second term on the right hand side. Observe that,

$$\mathop{\mathbb{E}}_{x}\left[\left(\mathop{\mathbb{E}}_{Z_r|x}\mathbb{1}_{|Z_r|>\alpha r}(e^{-\varepsilon Z_r/r^2} - 1)\right)\left(\mathop{\mathbb{E}}_{Z_r|x}\left[\frac{Z_r^2}{r^4}\right]\right)\right]^2 \le \mathop{\mathbb{E}}_{x}\left[\left(\mathop{\mathbb{E}}_{Z_r|x}\mathbb{1}_{|Z_r|>\alpha r}(e^{-\varepsilon Z_r/r^2} - 1)\right)^2\right]\mathop{\mathbb{E}}_{x}\left[\left(\mathop{\mathbb{E}}_{Z_r|x}\left[\frac{Z_r^2}{r^4}\right]\right)^2\right]$$

$$\le \mathop{\mathbb{E}}_{x}\left[\left(\mathop{\mathbb{E}}_{Z_r|x}\mathbb{1}_{|Z_r|>\alpha r}(e^{-\varepsilon Z_r/r^2} - 1)^2\right)\right]\mathop{\mathbb{E}}_{x}\left[\left(\mathop{\mathbb{E}}_{Z_r|x}\left[\frac{Z_r^4}{r^8}\right]\right)\right]$$

$$= \mathop{\mathbb{E}}_{Z_r}\left[\mathbb{1}_{|Z_r|>\alpha r}(e^{-\varepsilon Z_r/r^2} - 1)^2\right]\mathop{\mathbb{E}}_{Z_r}\left[\frac{Z_r^4}{r^8}\right]$$

$$\lesssim \frac{1}{r^4}\mathop{\mathbb{E}}_{Z_r}\left[\mathbb{1}_{|Z_r|>\alpha r}(e^{-\varepsilon Z_r/r^2} - 1)^2\right]$$

where the first inequality is by Cauchy-Schwarz, second inequality is by two applications of Jensen's inequality, and the last inequality is by properties of the Gaussian with standard deviation $r$.

We will now bound $\mathbb{E}_{Z_r}\left[\mathbb{1}_{|Z_r|>\alpha r}(e^{-\varepsilon Z_r/r^2} - 1)^2\right]$ in two separate regimes, when 1) $|Z_r| \le r^2/|\varepsilon|$ and when 2) $|Z_r| > r^2/|\varepsilon|$.

When $|Z_r| \le r^2/|\varepsilon|$, by linear approximations to the exponential function, we have

$$(e^{-\varepsilon Z_r/r^2} - 1)^2 \lesssim \frac{\varepsilon^2 Z_r^2}{r^4}$$

So,

$$\mathop{\mathbb{E}}_{Z_r}\left[\mathbb{1}_{|Z_r|>\alpha r}\mathbb{1}_{|Z_r|\le r^2/\varepsilon}(e^{-\varepsilon Z_r/r^2} - 1)^2\right] \lesssim \mathop{\mathbb{E}}_{Z_r}\left[\mathbb{1}_{|Z_r|>\alpha r}\frac{\varepsilon^2 Z_r^2}{r^4}\right] \lesssim \frac{\varepsilon^2\alpha^2}{r^2}e^{-\Omega(\alpha^2)}$$

On the other hand, when $|Z_r| > r^2/|\varepsilon|$,

$$\mathop{\mathbb{E}}_{Z_r}\left[\mathbb{1}_{|Z_r|>\max(\alpha r, r^2/\varepsilon)}(e^{-\varepsilon Z_r/r^2} - 1)^2\right]$$

$$= \int_{-\infty}^{-(\alpha r + r^2/\varepsilon)}\frac{1}{\sqrt{2\pi r^2}}e^{-\frac{z^2}{2r^2}}(e^{-\frac{\varepsilon z}{r^2}} - 1)^2\,\mathrm{d}z + \int_{\alpha r + r^2/\varepsilon}^{\infty}\frac{1}{\sqrt{2\pi r^2}}e^{-\frac{z^2}{2r^2}}(e^{-\frac{\varepsilon z}{r^2}} - 1)^2\,\mathrm{d}z$$

$$\lesssim \int_{\alpha r + r^2/\varepsilon}^{\infty}\frac{1}{\sqrt{2\pi r^2}}e^{-\frac{z^2}{2r^2}}(e^{\frac{|\varepsilon|z}{r^2}} - 1)^2\,\mathrm{d}z$$

$$\lesssim \int_{\alpha r + r^2/\varepsilon}^{\infty}\frac{1}{\sqrt{2\pi r^2}}e^{-\frac{z^2}{2r^2}}e^{\frac{2|\varepsilon|z}{r^2}}\,\mathrm{d}z$$

$$= e^{2\frac{\varepsilon^2}{r^2}}\int_{\alpha r + r^2/\varepsilon}^{\infty}\frac{1}{\sqrt{2\pi r^2}}e^{-\frac{(z - 2|\varepsilon|z)^2}{2r^2}}\,\mathrm{d}z$$

$$\lesssim e^{-\Omega(\alpha^2 + r^2/\varepsilon^2)} \lesssim \frac{\varepsilon^2}{r^2}e^{-\Omega(\alpha^2)} \quad \text{since } |\varepsilon| < r/2$$

Thus, we have shown that, whenever $\alpha \gtrsim 1$, we have

$$\mathbb{E}_x\left[\left(\mathbb{E}_{Z_r|x} \mathbb{1}_{|Z_r|>\alpha r}(e^{-\varepsilon Z_r/r^2}-1)\right)\left(\mathbb{E}_{Z_r|x}\left[\frac{Z_r}{r^2}\right]\right)\right] \lesssim \frac{1}{r^2}\sqrt{\frac{\varepsilon^2\alpha^2}{r^2}e^{-\Omega(\alpha^2)}+\frac{\varepsilon^2}{r^2}e^{-\Omega(\alpha^2)}}$$

$$\lesssim \frac{\varepsilon\alpha}{r^3}e^{-\Omega(\alpha^2)} \quad \text{since } \alpha \gtrsim 1$$

which implies that

$$\mathbb{E}_{x\sim f_r}\left[s_r^2(x+\varepsilon)\right] - \mathcal{I}_r \lesssim \frac{\varepsilon\alpha}{r}\left(\mathcal{I}_r + \frac{1}{r^2}e^{-\Omega(\alpha^2)}\right)$$

Set $\alpha = O(\sqrt{\log\frac{e}{r^2\mathcal{I}_r}})$, which is $\Omega(1)$ as $r^2\mathcal{I}_r \leq 1$ by Lemma 3.1. Therefore, we have

$$\mathbb{E}_{x\sim f_r}\left[s_r^2(x+\varepsilon)\right] - \mathcal{I}_r \lesssim \frac{\varepsilon}{r}\mathcal{I}_r\sqrt{\log\frac{e}{r^2\mathcal{I}_r}}$$

$\square$

With the above lemmas, we are now ready to prove Lemma 4.2, which we also restate here for the reader's convenience.

**Lemma 4.2.** *Suppose we have a known model $f_r$ that is the result of $r$-smoothing with Fisher information $\mathcal{I}_r$, and a given parameter $\varepsilon_{\max}$. Let $\beta$ and $\eta$ be the hidden multiplicative constants in Lemmas C.2 and C.3. Further suppose that $r$ satisfies $r \geq 2\varepsilon_{\max}$ and $r^2\sqrt{\mathcal{I}_r} \geq \gamma\varepsilon_{\max}$ for some parameter $\gamma \geq \beta$. Also define the notation $\tilde{\rho}$ (a "$o(1)$" term) by*

$$1+\tilde{\rho} = \sqrt{1+\frac{\eta\sqrt{\varepsilon}}{\gamma}} + \frac{15}{2\sqrt{\gamma}}\left(\frac{2\log\frac{1}{\delta}}{n}\right)^{\frac{1}{4}}$$

*Then, for every $\xi \ll 1$, with probability at least $1 - \delta \cdot \frac{2}{\xi r^2\mathcal{I}_r(1-\frac{\beta}{\gamma})(1-\delta)}$, for all $\varepsilon \in \left((1+\xi)\frac{1+\tilde{\rho}}{1-\frac{\beta}{\gamma}}\sqrt{\frac{2\log\frac{1}{\delta}}{n\mathcal{I}_r}}, \varepsilon_{\max}\right]$, $\hat{s}(\lambda-\varepsilon)$ is strictly negative and $\hat{s}(\lambda+\varepsilon)$ is strictly positive.*

*Proof.* Without loss of generality, we only show the $\lambda - \varepsilon$ case, and the $\lambda + \varepsilon$ case follows by doubling the failure probability.

First, we show that, under the lemma assumption that $r^2\sqrt{\mathcal{I}_r} \geq \gamma\varepsilon_{\max}$, Lemma C.3 implies that the second moment of the score at $\lambda-\varepsilon$, namely $\mathbb{E}_{x\sim f_r}[s_r^2(x+\varepsilon)]$, is upper bounded by $(1+O(1/\gamma))\mathcal{I}_r$.

To check the precondition of Lemma C.3, note that $r^2\sqrt{\mathcal{I}_r} \geq \gamma\varepsilon_{\max} \geq \gamma\varepsilon$ is equivalent to $r/\varepsilon \geq \gamma/\sqrt{r^2\mathcal{I}_r}$, which implies that

$$\frac{r}{\varepsilon} \geq \frac{\gamma}{\sqrt{r^2\mathcal{I}_r}}$$

$$= \frac{\gamma}{\sqrt{e}}\sqrt{\frac{e}{r^2\mathcal{I}_r}}$$

$$\geq \frac{\gamma}{\sqrt{e}}\sqrt{\log\frac{e}{r^2\mathcal{I}_r}}$$

satisfying the precondition of Lemma C.3.

Then, recalling the notation $\eta$ for the explicit constant in Lemma C.3, the lemma implies that

$$\mathbb{E}_{x \sim f_r} [s_r^2(x + \varepsilon)] \leq \mathcal{I}_r \left( 1 + \eta \frac{\varepsilon}{r} \sqrt{\log \frac{e}{r^2 \mathcal{I}_r}} \right)$$

$$\leq \mathcal{I}_r \left( 1 + \eta \frac{\varepsilon_{\max}}{r} \sqrt{\log \frac{e}{r^2 \mathcal{I}_r}} \right)$$

$$\leq \mathcal{I}_r \left( 1 + \eta \frac{\varepsilon_{\max}}{r} \sqrt{\frac{e}{r^2 \mathcal{I}_r}} \right)$$

$$= \mathcal{I}_r \left( 1 + \eta \sqrt{e} \frac{\varepsilon_{\max}}{r^2 \sqrt{\mathcal{I}_r}} \right)$$

$$\leq \mathcal{I}_r \left( 1 + \frac{\eta \sqrt{e}}{\gamma} \right)$$

Next, we show that with high probability, $\hat{s}(\lambda - \varepsilon)$ is upper bounded (by a quantity that we eventually show is negative, as desired). To do so, we use the concentration bound in Corollary 3.3, combined with the bound of Lemma C.2 on the expectation, as well the second moment bound for $\mathbb{E}_x[s_r^2(x + \varepsilon)]$ we just derived. Together, they imply that for all for all $0 < \varepsilon < \min(|r|, \varepsilon_{\max})$, with probability at least $1 - \delta$, we have

$$\hat{s}(\lambda - \varepsilon) - \left( -\mathcal{I}_r \varepsilon + \beta \sqrt{\mathcal{I}_r} \frac{\varepsilon^2}{r^2} \right) \leq \sqrt{1 + \frac{\eta \sqrt{e}}{\gamma}} \sqrt{\frac{2 \log \frac{1}{\delta}}{n} \mathcal{I}_r} + \frac{15 \log \frac{1}{\delta}}{nr}$$

We will show that the last term of $O(\log \frac{1}{\delta}/(nr))$ is dominated by the first term, by our lemma assumption. Note that since $r^2 \sqrt{\mathcal{I}_r} \geq \gamma \varepsilon_{\max} \geq \gamma \sqrt{\frac{2 \log \frac{1}{\delta}}{n} \frac{1}{\mathcal{I}_r}}$, we can bound the above by

$$\hat{s}(\lambda - \varepsilon) - \left( -\mathcal{I}_r \varepsilon + \beta \sqrt{\mathcal{I}_r} \frac{\varepsilon^2}{r^2} \right) \leq \sqrt{1 + \frac{\eta \sqrt{e}}{\gamma}} \sqrt{\frac{2 \log \frac{1}{\delta}}{n} \mathcal{I}_r} + \frac{15}{2\sqrt{\gamma}} \left( \frac{2 \log \frac{1}{\delta}}{n} \right)^{\frac{1}{4}} \sqrt{\frac{2 \log \frac{1}{\delta}}{n} \mathcal{I}_r}$$

$$= \left( \sqrt{1 + \frac{\eta \sqrt{e}}{\gamma}} + \frac{15}{2\sqrt{\gamma}} \left( \frac{2 \log \frac{1}{\delta}}{n} \right)^{\frac{1}{4}} \right) \sqrt{\frac{2 \log \frac{1}{\delta}}{n} \mathcal{I}_r}$$

For the rest of the proof, we denote the multiplicative term $\sqrt{1 + \frac{\eta \sqrt{e}}{\gamma}} + \frac{15}{2\sqrt{\gamma}} \left( \frac{2 \log \frac{1}{\delta}}{n} \right)^{\frac{1}{4}}$ simply by $1 + \tilde{\rho}$, as defined in the theorem statement.

We note also that, since $r^2 \sqrt{\mathcal{I}_r} \geq \gamma \varepsilon$, we have $-\mathcal{I}_r \varepsilon + \beta \sqrt{\mathcal{I}_r} \frac{\varepsilon^2}{r^2} \leq \left( -1 + \frac{\beta}{\gamma} \right) \mathcal{I}_r \varepsilon$. Therefore, for any $0 < \varepsilon < \min(|r|, \varepsilon_{\max})$, we have that with probability at least $1 - \delta$,

$$\hat{s}(\lambda - \varepsilon) \leq \left( -1 + \frac{\beta}{\gamma} \right) \mathcal{I}_r \varepsilon + (1 + \tilde{\rho}) \sqrt{\frac{2 \log \frac{1}{\delta}}{n} \mathcal{I}_r}$$

By Lemma B.2, we also have that for *any* $x$

$$\hat{s}'(x) \leq \frac{1}{r^2}$$

Let $\xi \ll 1$ be a parameter that we choose at the end of the proof. We will show that with probability at least $1 - \delta \cdot \frac{1}{\xi r^2 \mathcal{I}_r (1 - \frac{\beta}{\gamma})(1 - \delta)}$, we have for all $\varepsilon \in \left( (1 + \xi) \frac{1 + \tilde{\rho}}{1 - \frac{\beta}{\gamma}} \sqrt{\frac{2 \log \frac{1}{\delta}}{n \mathcal{I}_r}}, \varepsilon_{\max} \right]$, $\hat{s}(\lambda - \varepsilon) < 0$.

Consider a net $N$ of spacing $\xi r^2 (1 + \tilde{\rho}) \sqrt{\frac{2 \log \frac{1}{\delta}}{n} \mathcal{I}_r}$ over the interval $\left( (1 + \xi) \frac{1 + \tilde{\rho}}{1 - \frac{\beta}{\gamma}} \sqrt{\frac{2 \log \frac{1}{\delta}}{n \mathcal{I}_r}}, \varepsilon_{\max} \right]$ in the theorem statement. We can check that, if for all points $\varepsilon \in N$, we have

$$\hat{s}(\lambda - \varepsilon) \leq -\xi (1 + \tilde{\rho}) \sqrt{\frac{2 \log \frac{1}{\delta}}{n} \mathcal{I}_r} \tag{8}$$

then, because $\hat{s}' \leq 1/r^2$, we have $\hat{s}(x) < 0$ for all $x \in \lambda + \left( (1+\xi)\frac{1+\tilde{\rho}}{1-\frac{\beta}{\gamma}}\sqrt{\frac{2\log\frac{1}{\delta}}{n\mathcal{I}_r}}, \varepsilon_{\max} \right]$. This is done by considering two consecutive net points $0 < \varepsilon_1 < \varepsilon_2$, and observing that $\hat{s}(\lambda - \varepsilon) \leq \hat{s}(\lambda - \varepsilon_1) - \frac{\varepsilon_1 - \varepsilon}{r^2}$ for $\varepsilon \in (\varepsilon_1, \varepsilon_2]$, which is in turn strictly negative. (For the essentially symmetric case of $\lambda + \varepsilon$, we would instead use the inequality that $\hat{s}(\lambda + \varepsilon) \geq \hat{s}(\lambda + \varepsilon_1) + \frac{\varepsilon_1 - \varepsilon}{r^2} > 0$.)

Thus, it suffices to bound the probability that the above inequality holds for all points in $N$. For a natural number $i \geq 1$, consider the subset $N_i$ of $N$ that intersects with $([i, i+1] + \xi)\frac{1+\tilde{\rho}}{1-\frac{\beta}{\gamma}}\sqrt{\frac{2\log\frac{1}{\delta}}{n\mathcal{I}_r}}$, where here we interpret addition and multiplication as scalar operations on every point in the interval. For each $\varepsilon \in N_i$, Equation 8 holds except for probability $\delta^{i^2}$. Furthermore, each $N_i$ consists of $\frac{1+\tilde{\rho}}{1-\frac{\beta}{\gamma}}\sqrt{\frac{2\log\frac{1}{\delta}}{n\mathcal{I}_r}}$ divided by $\xi r^2(1+\tilde{\rho})\sqrt{\frac{2\log\frac{1}{\delta}}{n}}\mathcal{I}_r$ many points, which equals to $\frac{1}{\xi r^2 \mathcal{I}_r(1-\frac{\beta}{\gamma})}$ many points. Therefore, the total failure probability is at most $\frac{1}{\xi r^2 \mathcal{I}_r(1-\frac{\beta}{\gamma})} \cdot \sum_{i \geq 1} \delta^{i^2} \leq \delta \cdot \frac{1}{\xi r^2 \mathcal{I}_r(1-\frac{\beta}{\gamma})(1-\delta)}$. An extra factor of 2 in the failure probability in the theorem statement accounts for the symmetric case of $\hat{s}(\lambda + \varepsilon) > 0$. $\qquad \square$

# D Proof of Theorem 1.5 in Section 6

The goal of this appendix is to prove Theorem 1.5, which we restate here for the reader's convenience.

**Theorem 1.5.** *Suppose $f_r$ is an $r$-smoothed distribution with Fisher information $\mathcal{I}_r$. Given failure probability $\delta$ and sample size $n$, no algorithm can distinguish $f_r$ and $f_r^{2\varepsilon}$ with probability $1 - \delta$, where $\varepsilon = (1 - o(1))\sqrt{2\log\frac{1}{\delta}/(n\mathcal{I}_r)}$. Here, the $o(1)$ term tends to 0 as $\delta \to 0$ and $\log\frac{1}{\delta}/n \to 0$, for a fixed $r^2\mathcal{I}_r$.*

We use the standard proof technique of reducing distinguishing two "close" distributions to estimation. In particular, we show that it is statistically impossible to distinguish between $f_r$ and $f_r^{2\varepsilon}$ with probability $1 - \delta$ using $n$ samples. In order to show such an indistinguishability result, we need the following standard fact (essentially the Neyman-Pearson lemma):

**Fact D.1.** *Consider a game, where an adversary picks arbitrarily either distribution $p$ or distribution $q$, and we want an algorithm which, on input $n$ independent samples from the chosen distribution, decide whether the samples came from $p$ or $q$, succeeding with probability at least $1 - \delta$. Then, there is no algorithm $\mathcal{A}$ such that:*

$$\mathbb{P}(\mathcal{A} \text{ returns } p \mid \text{adversary picked } p) - \mathbb{P}(\mathcal{A} \text{ returns } p \mid \text{adversary picked } q) > d_{\mathrm{TV}}(p^{\otimes n}, q^{\otimes n})$$

*where $p^{\otimes n}$ denotes the $n$-fold product distribution of $p$. In particular, this implies that there is no algorithm $\mathcal{A}$ such that both of the following hold:*

- $\mathbb{P}(\mathcal{A} \text{ returns } p \mid \text{adversary picked } p) > \frac{1}{2} + \frac{1}{2}d_{\mathrm{TV}}(p^{\otimes n}, q^{\otimes n})$
- $\mathbb{P}(\mathcal{A} \text{ returns } q \mid \text{adversary picked } q) > \frac{1}{2} + \frac{1}{2}d_{\mathrm{TV}}(p^{\otimes n}, q^{\otimes n})$

*So if $d_{\mathrm{TV}}(p^{\otimes n}, q^{\otimes n}) < 1 - 2\delta$, there is no algorithm that will succeed in distinguishing between two distributions with probability $\geq 1 - \delta$ using only $n$ samples.*

Thus, we need to upper bound the $n$-sample total variation distance between $f_r$ and $f_r^{2\varepsilon}$. Standard inequalities for doing so involve calculating and plugging-in the single-sample KL-divergence $D_{\mathrm{KL}}(f_r \parallel f_r^{2\varepsilon})$ or squared Hellinger distance $d_{\mathrm{H}}^2(f_r, f_r^{2\varepsilon})$, however, they yield only constant factor tightness in the exponent of $1 - d_{\mathrm{TV}}(p^{\otimes n}, q^{\otimes n})$, and hence only constant factor tightness in sample complexity or estimation error lower bounds. As such, in this paper, we prove a new lemma (Lemma D.2) that involves both the KL-divergence and squared Hellinger distance, as well as assumptions on the concentration of the log-likelihood ratio between $f_r$ and $f_r^{2\varepsilon}$ (which will be satisfied by $r$-smoothed distributions), which allows us to bound the $n$-sample total variation distance tightly. After that, we calculate the KL divergence and squared Hellinger distance of $f_r$ and $f_r^{2\varepsilon}$ as well as show the concentration of their log likelihood ratio (Appendix D.1), which when applied to the lemma yields the lower bound result (Appendix D.2).

**Lemma D.2.** *Consider two arbitrary distributions $p$, $q$. Let the log-likelihood ratio be defined as $\gamma = \log \frac{q}{p}$. Suppose there is a parameter $\kappa > 0$ such that we have the following conditions:*

1. $d_{\mathrm{H}}^2(p, q) \leq (1 + \kappa)\frac{1}{4} D_{\mathrm{KL}}(p \,\|\, q)$

2. $\frac{D_{\mathrm{KL}}(p \,\|\, q)}{D_{\mathrm{KL}}(q \,\|\, p)} \in [(1 + \kappa)^{-1}, 1 + \kappa]$

3. $\mathbb{E}_p[|\gamma|^k] \leq (1 + \kappa)\frac{k!}{2} 2 D_{\mathrm{KL}}(p \,\|\, q)\kappa^{k-2}$ *for integers $k \geq 2$*

4. $\mathbb{E}_q[|\gamma|^k] \leq (1 + \kappa)\frac{k!}{2} 2 D_{\mathrm{KL}}(p \,\|\, q)\kappa^{k-2}$ *for integers $k \geq 2$*

*Then, for $\kappa \leq 0.01$, $n D_{\mathrm{KL}}(p \,\|\, q) \gg 1$ and $D_{\mathrm{KL}}(p \,\|\, q) \ll 1$,*

$$1 - d_{\mathrm{TV}}(p^{\otimes n}, q^{\otimes n}) \geq 2 e^{-(1 + O(\kappa) + O(1/\sqrt{n D_{\mathrm{KL}}(p \,\|\, q)}) + O(D_{\mathrm{KL}}(p \,\|\, q)))n D_{\mathrm{KL}}(p \,\|\, q)/4}$$

*Proof.* Define

$$BC_S(p, q) = \int_S \sqrt{pq} \leq \sqrt{p(S)q(S)}$$

to be the restriction of the Bhattacharyya coefficient $BC(p, q) = 1 - d_{\mathrm{H}}^2(p, q)$ to a subset $S$ of the domain. For any $S$, we have

$$1 - d_{\mathrm{TV}}(p, q) = \int \min(p, q) \geq \int_S \min(p, q) \geq \frac{(\int_S \sqrt{\min(p, q)\max(p, q)})^2}{\int_S \max(p, q)} = \frac{BC_S(p, q)^2}{p(S) + q(S)}.$$

We apply this to $p^{\otimes n}$ and $q^{\otimes n}$, getting for any $S$:

$$1 - d_{\mathrm{TV}}(p^{\otimes n}, q^{\otimes n}) \geq \frac{BC_S(p^{\otimes n}, q^{\otimes n})^2}{p^{\otimes n}(S) + q^{\otimes n}(S)}. \tag{9}$$

Thus, the goal now is to find an event $S$ such that $BC_S(p^{\otimes n}, q^{\otimes n})$ is big relative to $p^{\otimes n}(S) + q^{\otimes n}(S)$.

For the rest of the proof, we use the notation $\bar{\gamma}$ to denote the $n$-sample empirical log-likelihood ratio, namely $\frac{1}{n} \sum_i \gamma_i = \frac{1}{n} \sum_i \log \frac{q(x_i)}{p(x_i)}$.

We now define $S_k$, for $k \in \mathbb{Z}$, to be the event $\{\bar{\gamma} \in [k - \frac{1}{2}, k + \frac{1}{2}] \cdot \alpha D_{\mathrm{KL}}(p \,\|\, q)\}$ for some parameter $\alpha = \Theta(\max(\kappa, 1/\sqrt{n D_{\mathrm{KL}}(p \,\|\, q)}, D_{\mathrm{KL}}(p \,\|\, q)))$, and set $S = S_0$. We have that

$$BC_S(p^{\otimes n}, q^{\otimes n}) = BC(p^{\otimes n}, q^{\otimes n}) - \sum_{k \neq 0} BC_{S_k}(p^{\otimes n}, q^{\otimes n}) \geq BC(p, q)^n - \sum_{k \neq 0} \sqrt{p^{\otimes n}(S_k)q^{\otimes n}(S_k)} \tag{10}$$

Now, define

$$\delta := e^{-n \min(D_{\mathrm{KL}}(p \,\|\, q), D_{\mathrm{KL}}(q \,\|\, p))/4}$$

We note that $\delta \leq (BC(p, q)^n)^{(1 - O(\kappa) - O(D_{\mathrm{KL}}(p \,\|\, q)))}$, as follows:

$$BC(p, q) = 1 - d_{\mathrm{H}}^2(p, q)$$

$$\geq 1 - (1 + O(\kappa))\frac{1}{4}\min(D_{\mathrm{KL}}(p \,\|\, q), D_{\mathrm{KL}}(q \,\|\, p))$$

$$\geq \exp\left(-(1 + O(\kappa) + O(D_{\mathrm{KL}}(p \,\|\, q)))\frac{1}{4}\min(D_{\mathrm{KL}}(p \,\|\, q), D_{\mathrm{KL}}(q \,\|\, p))\right)$$

where the first inequality follows from conditions 1 and 2 in the lemma statement, and the second inequality follows from the fact that $1 - x = \exp(-(1 + \Theta(x))x)$. The above claim follows from raising both sides to the power of $n$.

We shall now bound $p^{\otimes n}(S_k)$ and $q^{\otimes n}(S_k)$ in terms of $\delta$. By standard sub-Gamma concentration bounds, conditions 3 and 4 imply that

$$\Pr_p\left[\bar{\gamma} > -D_{\mathrm{KL}}(p \,\|\, q) + t\sqrt{2(1 + \kappa)D_{\mathrm{KL}}(p \,\|\, q)} + \frac{\kappa}{2}t^2\right] < e^{-nt^2/2} \tag{11}$$

and

$$\Pr_q \left[ \overline{\gamma} < D_{\mathrm{KL}}(q \,\|\, p) - t\sqrt{2(1+\kappa)D_{\mathrm{KL}}(p \,\|\, q)} - \frac{\kappa}{2}t^2 \right] < e^{-nt^2/2} \tag{12}$$

We now bound $p^{\otimes n}(S_k)$ by

$$p^{\otimes n}(S_k) \le \Pr_p \left[ \overline{\gamma} \ge \left( k - \frac{1}{2} \right) \alpha D_{\mathrm{KL}}(p \,\|\, q) \right]$$

Solving the equation

$$\frac{\kappa}{2}t_k^2 + \sqrt{2(1+\kappa)D_{\mathrm{KL}}(p \,\|\, q)}\, t_k - D_{\mathrm{KL}}(p \,\|\, q) = \left( k - \frac{1}{2} \right) \alpha D_{\mathrm{KL}}(p \,\|\, q)$$

yields

$$t_k = \frac{\sqrt{1+\kappa}}{\kappa} \sqrt{2D_{\mathrm{KL}}(p \,\|\, q)} \left( \sqrt{1 + \frac{\kappa}{1+\kappa} \left( 1 + \left( k - \frac{1}{2} \right) \alpha \right)} - 1 \right)$$

By Equation 11, $p^{\otimes n}(S_k) \le e^{-nt_k^2/2}$ whenever $(k - \frac{1}{2})\alpha > -1$.

Also observe that, when $\kappa \le 0.01$, the function $\frac{1}{\kappa}(\sqrt{1 + \frac{\kappa}{1+\kappa}(1+x)} - 1)$ within the range $x \in [-1, 1.01]$ can be lower bounded by simply $\frac{1}{2}(1 - 2\kappa)(1 + x)$. For the range $x \ge 1$, we can lower bound the function by $(1 - 2\kappa)\sqrt{x}$. This implies that $p^{\otimes n}(S_k)$ can be upper bounded by $e^{-\frac{n}{4}(1-O(\kappa))D_{\mathrm{KL}}(p \,\|\, q)(1+(k-\frac{1}{2})\alpha)^2} \le \delta^{(1-O(\kappa))(1+(k-\frac{1}{2})\alpha)^2}$ when $(k - \frac{1}{2})\alpha \in [-1, 1.01]$, and similarly, upper bounded by $e^{-n(1-O(\kappa))D_{\mathrm{KL}}(p \,\|\, q)(k-\frac{1}{2})\alpha} \le \delta^{4(1-O(\kappa))(k-\frac{1}{2})\alpha}$ when $(k - \frac{1}{2})\alpha \ge 1$. Finally, observe that for $(k - \frac{1}{2})\alpha \le -1$, we can trivially bound $p^{\otimes n}(S_k)$ by 1.

We now bound $q^{\otimes n}(S_k)$ by

$$q^{\otimes n}(S_k) \le \Pr_q \left[ \overline{\gamma} \le \left( k + \frac{1}{2} \right) \alpha D_{\mathrm{KL}}(p \,\|\, q) \right]$$

Solving the equation

$$-\frac{\kappa}{2}(t_k')^2 - \sqrt{2(1+\kappa)D_{\mathrm{KL}}(p \,\|\, q)}\, t_k' + D_{\mathrm{KL}}(q \,\|\, p) = \left( k + \frac{1}{2} \right) \alpha D_{\mathrm{KL}}(p \,\|\, q)$$

yields (by condition 2)

$$t_k' \ge \frac{\sqrt{1+\kappa}}{\kappa} \sqrt{2D_{\mathrm{KL}}(p \,\|\, q)} \left( \sqrt{1 + \frac{\kappa}{1+\kappa} \left( 1 - \kappa - \left( k + \frac{1}{2} \right) \alpha \right)} - 1 \right)$$

By Equation 12, $q^{\otimes n}(S_k) \le e^{-n(t_k')^2/2}$ whenever $(k + \frac{1}{2})\alpha < 1 - \kappa$.

We now bound $q^{\otimes n}(S_k)$ similar to how we bounded $p^{\otimes n}(S_k)$. When $\kappa \le 0.01$, the function $\frac{1}{\kappa}(\sqrt{1 + \frac{\kappa}{1+\kappa}(1 - \kappa + x)} - 1)$ within the range $x \in [-1.01, 1 - \kappa]$ can be lower bounded by simply $\frac{1}{2}(1 - 2\kappa)(1 - \kappa - x)$. For the range $x \le -1$, we can lower bound the function by $(1 - 2\kappa)\sqrt{x}$. This implies that $q^{\otimes n}(S_k)$ can be upper bounded by $e^{-\frac{n}{4}(1-O(\kappa))D_{\mathrm{KL}}(p \,\|\, q)(1 - \kappa - (k+\frac{1}{2})\alpha)^2} \le \delta^{(1-O(\kappa))(1 - \kappa - (k+\frac{1}{2})\alpha)^2}$ when $(k + \frac{1}{2})\alpha \in [-1.01, 1 - \kappa]$, and similarly, upper bounded by $e^{n(1-O(\kappa))D_{\mathrm{KL}}(p \,\|\, q)(k+\frac{1}{2})\alpha} \le \delta^{-4(1-O(\kappa))(k+\frac{1}{2})\alpha}$ when $(k + \frac{1}{2})\alpha \le -1$. Finally, observe that for $(k + \frac{1}{2})\alpha \ge 1 - \kappa$, we can trivially bound $q^{\otimes n}(S_k)$ by 1.

We are now ready to upper bound $\sum_{k \ne 0} \sqrt{p^{\otimes n}(S_k)q^{\otimes n}(S_k)}$. We decompose this sum into three regions of non-zero $k$.

The main region $K_1$ is where $(k - \frac{1}{2})\alpha \ge -1$ and $(k + \frac{1}{2})\alpha \le 1 - \kappa$. In this region, $\sqrt{p^{\otimes n}(S_k)q^{\otimes n}(S_k)}$ can be upper bounded by $\delta^{\frac{1-O(\kappa)}{2}\left[(1+(k-\frac{1}{2})\alpha)^2 + (1 - \kappa - (k+\frac{1}{2})\alpha)^2\right]} \le \delta^{(1-O(\kappa))(1+(|k|-\frac{1}{2})^2\alpha^2))}$. Now observe that $\sum_{k \in K_1} \delta^{(1-O(\kappa))(1+(|k|-\frac{1}{2})^2\alpha^2))} \lesssim \delta^{1-O(\kappa)+\Omega(\alpha)} \le$

$\delta^{1+\Omega(\alpha)}$ as long as $\delta^{O(\alpha^2)} \ll 1$ and $\alpha = \Omega(\kappa)$. These two conditions are satisfied by the choice of $\alpha = \Omega(\max(\kappa, 1/\sqrt{nD_{\mathrm{KL}}(p \,\|\, q)}))$.

The second region $K_2$ is where $(k - \frac{1}{2})\alpha \leq -1$, which also means that $(k + \frac{1}{2})\alpha \leq -(1 + \Omega(\alpha))$. In this case, we use the bound $p^{\otimes n}(S_k) \leq 1$ and $q^{\otimes n}(S_k) \leq \delta^{-4(1-O(\kappa))(k+\frac{1}{2})\alpha}$. Thus, in this region, $\sqrt{p^{\otimes n}(S_k)q^{\otimes n}(S_k)}$ is upper bounded $\delta^{-2(1-O(\kappa))(k+\frac{1}{2})\alpha}$. This means that $\sum_{k \in K_2} \sqrt{p^{\otimes n}(S_k)q^{\otimes n}(S_k)} \lesssim \delta^{2(1-O(\kappa))(1-O(\alpha))} \ll \delta^{1+\Omega(\alpha)}$, as long as $\delta^\alpha \ll 1$ and as long as $\kappa \ll 1$ and $\alpha \ll 1$. This is again satisfied by our choice of $\alpha$.

The last region $K_3$ is where $(k + \frac{1}{2})\alpha \geq 1 - \kappa$, which also means that $(k - \frac{1}{2})\alpha \geq (1 - O(\kappa) - O(\alpha))$. In this case, we use the bound $p^{\otimes n}(S_k) \leq \delta^{4(1-O(\kappa))(k-\frac{1}{2})\alpha}$ and $q^{\otimes n}(S_k) \leq 1$. Thus, in this region, $\sqrt{p^{\otimes n}(S_k)q^{\otimes n}(S_k)}$ is upper bounded $\delta^{2(1-O(\kappa))(k-\frac{1}{2})\alpha}$. This means that $\sum_{k \in K_3} \sqrt{p^{\otimes n}(S_k)q^{\otimes n}(S_k)} \lesssim \delta^{2(1-O(\kappa))(1-O(\alpha))} \ll \delta^{1+\Omega(\alpha)}$, as long as $\delta^\alpha \ll 1$ and as long as $\kappa \ll 1$ and $\alpha \ll 1$. This is again satisfied by our choice of $\alpha$.

Summarizing, we have shown that $\sum_{k \neq 0} \sqrt{p^{\otimes n}(S_k)q^{\otimes n}(S_k)} \lesssim \delta^{1+\Omega(\alpha)}$ for $\alpha = \Omega(\max(\kappa, 1/\sqrt{nD_{\mathrm{KL}}(p \,\|\, q)}))$. Furthermore, as $\alpha \geq \Omega(D_{\mathrm{KL}}(p \,\|\, q))$ by construction, the above bound is much less than $(BC(p, q))^n$, since $(BC(p, q))^n \geq \delta^{1+O(\kappa)+O(D_{\mathrm{KL}}(p \,\|\, q))}$ from earlier in this proof. This yields that $BC_S(p^{\otimes n}, q^{\otimes n}) \geq BC(p, q)^n - \sum_{k \neq 0} \sqrt{p^{\otimes n}(S_k)q^{\otimes n}(S_k)} \geq \delta^{1+O(\alpha)}$ since $\alpha = \Omega(D_{\mathrm{KL}}(p \,\|\, q))$.

The last quantities we have to bound are $p^{\otimes n}(S_0)$ and $q^{\otimes n}(S_0)$. These were already bounded in the respective paragraphs bounding $p^{\otimes n}(S_k)$ and $q^{\otimes n}(S_k)$ for general $k$. When $k = 0$, the bounds are at most $\delta^{1+\Omega(\alpha)}$ (again, when $\alpha = \Omega(\kappa)$). Finally, we get that

$$1 - d_{\mathrm{TV}}(p^{\otimes n}, q^{\otimes n}) \geq \frac{(BC_S(p^{\otimes n}, q^{\otimes n}))^2}{p^{\otimes n}(S) + q^{\otimes n}(S)} \gtrsim \frac{\delta^{2+O(\alpha)}}{\delta^{1+\Omega(\alpha)}} = \delta^{1+O(\alpha)}$$

Expanding the definition of $\delta$ as well as the choice of $\alpha = \Theta(\max(\kappa, 1/\sqrt{nD_{\mathrm{KL}}(p \,\|\, q)}, D_{\mathrm{KL}}(p \,\|\, q)))$ gives the lemma statement. $\qquad \square$

## D.1 Showing the conditions for Lemma D.2

In this subsection, we calculate the KL-divergence, squared Hellinger distance, as well as moment bounds for the log-likelihood ratio for $f_r$ and $f_r^{2\varepsilon}$ for a generic $r$-smoothed distribution $f_r$.

**Lemma D.3.** *Consider the parametric family $f_r^\lambda(x) = f_r(x - \lambda)$ for some $r$-smoothed distribution $f_r$ with Fisher information $\mathcal{I}_r$. Then for $\varepsilon < \frac{r}{4}$,*

$$D_{\mathrm{KL}}(f_r \,\|\, f_r^{2\varepsilon}) = 2\varepsilon^2 \mathcal{I}_r \left( 1 + \Theta\left( \frac{\varepsilon}{r^2\sqrt{\mathcal{I}_r}} \right) \right)$$

*Proof.* Let $\ell(x) = \log f_r(x)$

$$D_{\mathrm{KL}}(f_r \,\|\, f_r^{2\varepsilon}) = -\int_{-\infty}^{\infty} f_r(x) \log \frac{f_r^{2\varepsilon}(x)}{f_r(x)} \, \mathrm{d}x$$

$$= -\int_{-\infty}^{\infty} f_0(x) \int_x^{x+2\varepsilon} \ell'(y) \, \mathrm{d}y$$

$$= -\int_0^{2\varepsilon} \mathbb{E}_{z \leftarrow f_r} [s_r(z + y)] \, \mathrm{d}y$$

$$= \int_0^{2\varepsilon} \mathcal{I}_r y + \Theta\left( \sqrt{\mathcal{I}_r} \frac{y^2}{r^2} \right) \mathrm{d}y$$

$$= 2\varepsilon^2 \mathcal{I}_r + \Theta\left( \sqrt{\mathcal{I}_r} \frac{\varepsilon^3}{r^2} \right) = 2\varepsilon^2 \mathcal{I}_r \left( 1 + \Theta\left( \frac{\varepsilon}{r^2\sqrt{\mathcal{I}_r}} \right) \right)$$

where the $\Theta$ result is from Lemma C.2.

$\qquad \square$

**Lemma D.4.** *Consider the parametric family $f_r^\lambda(x) = f_r(x - \lambda)$ for some $r$-smoothed distribution $f_r$ with Fisher information $\mathcal{I}_r$. Then for $\varepsilon \leq r$,*

$$d_{\mathrm{H}}^2(f_r, f_r^{2\varepsilon}) \leq \frac{1}{4} D_{\mathrm{KL}}(f_r \parallel f_r^{2\varepsilon}) + O\left(\frac{\varepsilon^3}{r^3}\right)$$

*Proof.* Observe that the squared Hellinger distance is

$$d_{\mathrm{H}}^2(f_r, f_r^{2\varepsilon}) = \frac{1}{2} \int_{-\infty}^{\infty} (\sqrt{f_r(x)} - \sqrt{f_r^{2\varepsilon}(x)})^2 \, \mathrm{d}x$$

$$= \frac{1}{2} \int_{-\infty}^{\infty} f_r(x)(1 - \sqrt{\frac{f_r^{2\varepsilon}(x)}{f_r(x)}})^2 \, \mathrm{d}x$$

Since for $y > 0$ we have $(1 - \sqrt{y})^2 \leq -\frac{1}{2}(\log y) + \frac{y-1}{2} + \frac{1}{24}(y-1)_+^3$, substituting $y = \frac{f_r^{2\varepsilon}(x)}{f_r(x)}$, the previous expression is at most

$$\frac{1}{2} \int_{-\infty}^{\infty} f_r(x) \left( -\frac{1}{2} \log \frac{f_r^{2\varepsilon}(x)}{f_r(x)} + \frac{1}{2} \frac{f_r^{2\varepsilon}(x)}{f_r(x)} - \frac{1}{2} + \frac{1}{24} \left( \frac{f_r^{2\varepsilon}(x)}{f_r(x)} - 1 \right)_+^3 \right) dx$$

The first term inside the big parentheses equals $\frac{1}{4} D_{\mathrm{KL}}(f_r \parallel f_r^{2\varepsilon})$; the next two terms cancel out (since they each integrate all the probability mass of $f$); the final, cubic, term we bound now.

We start by bounding the cubic term for a Gaussian $g$ of standard deviation $r$:

$$\int_{-\infty}^{\infty} g(x) \left( \frac{g(x + 2\varepsilon)}{g(x)} - 1 \right)_+^3 \, \mathrm{d}x = O\left(\frac{\varepsilon^3}{r^3}\right)$$

where the bound is easily computed from the closed form evaluation of the integral, valid while $\varepsilon$ is bounded by some fixed multiple of $r$.

Now the $r$-smoothed distribution $f$ is just a convex combination of Gaussians of width $r$, and the desired inequality follows from the observation that the expression $f_r(x) \left( \frac{f_r^{2\varepsilon}(x)}{f_r(x)} - 1 \right)_+^3$ is convex in the sense that, in terms of $y, z > 0$, the function $y \left( \frac{z}{y} - 1 \right)_+^3$ is a convex 2-variable function. Thus the total contribution of the cubic term is bounded by its value for the Gaussian, namely $O(\frac{\varepsilon^3}{r^3})$. $\qquad\square$

**Lemma D.5.** *Let $k \geq 3$. For an $r$-smoothed distribution $f_r$, let $\gamma = \log \frac{f_r^{2\varepsilon}}{f_r}$. For $\varepsilon \leq r$, we have*

$$\mathbb{E}_p[|\gamma|^k] \leq \frac{k!}{2}(30\varepsilon/r)^{k-2} 4\varepsilon^2 \mathcal{I} \left( 1 + O\left( \frac{\varepsilon}{r} \sqrt{\log \frac{1}{r^2 \mathcal{I}}} \right) \right)$$

*Proof.* Let $\ell(x) = \log f_r(x)$. We have

$$\int_{-\infty}^{\infty} f_r(x) \left|\log f_r(x) - \log f_r(x + 2\varepsilon)\right|^k = \int_{-\infty}^{\infty} f_r(x) \left|\int_x^{x+2\varepsilon} \ell'(y)\,\mathrm{d}y\right|^k \mathrm{d}x$$

$$\leq (2\varepsilon)^{k-1} \int_{-\infty}^{\infty} f_r(x) \int_x^{x+2\varepsilon} |\ell'(y)|^k \,\mathrm{d}y\,\mathrm{d}x$$

$$= (2\varepsilon)^{k-1} \int_0^{2\varepsilon} \mathbb{E}_x[|s_r(x+y)|^k]\,\mathrm{d}y$$

$$\leq (2\varepsilon)^{k-1} \frac{k!}{2}(15/r)^{k-2} \int_0^{2\varepsilon} \mathbb{E}_x[s_r(x+y)^2]\,\mathrm{d}y$$

$$\leq (2\varepsilon)^{k-1} \frac{k!}{2}(15/r)^{k-2} \int_0^{2\varepsilon} \mathcal{I} + O\left(\frac{y}{r}\mathcal{I}\sqrt{\log \frac{1}{r^2\mathcal{I}}}\right)\,\mathrm{d}y$$

$$= (2\varepsilon)^{k} \frac{k!}{2}(15/r)^{k-2}\mathcal{I}\left(1 + O\left(\frac{\varepsilon}{r}\sqrt{\log \frac{1}{r^2\mathcal{I}}}\right)\right)$$

$$= \frac{k!}{2}(30\varepsilon/r)^{k-2}4\varepsilon^2\mathcal{I}\left(1 + O\left(\frac{\varepsilon}{r}\sqrt{\log \frac{1}{r^2\mathcal{I}}}\right)\right)$$

where the first three inequalities are by convexity, by Lemma B.6, and by Lemma C.3. □

**Lemma D.6.** *For an $r$-smoothed distribution $f_r$, let $\gamma = \log \frac{f_r^{2\varepsilon}}{f_r}$. For $\varepsilon \leq r$, we have*

$$\mathbb{E}_{f_r}[|\gamma|^2] \leq 4\varepsilon^2\mathcal{I}\left(1 + O\left(\frac{\varepsilon}{r}\sqrt{\log \frac{1}{r^2\mathcal{I}}}\right)\right)$$

*Proof.* Let $\ell(x) = \log f_r(x)$. We have

$$\int_{-\infty}^{\infty} f_r(x) \left|\log f_r(x) - \log f_r(x + 2\varepsilon)\right|^2 = \int_{-\infty}^{\infty} f_r(x) \left|\int_x^{x+2\varepsilon} \ell'(y)\,\mathrm{d}y\right|^2 \mathrm{d}x$$

$$\leq 2\varepsilon \int_{-\infty}^{\infty} f_r(x) \int_x^{x+2\varepsilon} |\ell'(y)|^2 \,\mathrm{d}y\,\mathrm{d}x$$

$$= 2\varepsilon \int_0^{2\varepsilon} \mathbb{E}_x[s_r(x+y)^2]\,\mathrm{d}y$$

$$\leq 2\varepsilon \int_0^{2\varepsilon} \mathcal{I} + O\left(\frac{y}{r}\mathcal{I}\sqrt{\log \frac{1}{r^2\mathcal{I}}}\right)\,\mathrm{d}y$$

$$= 4\varepsilon^2\mathcal{I}\left(1 + O\left(\frac{\varepsilon}{r}\sqrt{\log \frac{1}{r^2\mathcal{I}}}\right)\right)$$

where the first two inequalities are by convexity, and by Lemma C.3. □

### D.2 Proving Theorem 1.5

We are ready to prove Theorem 1.5.

*Proof of Theorem 1.5.* We will be applying Lemma D.2 on the distributions $f_r$ and $f_r^{2\varepsilon}$ for an appropriately chosen $\varepsilon$, with $\kappa = O(\frac{\varepsilon}{r}\frac{1}{r^2\mathcal{I}_r})$ (note that by Lemma 3.1, $\mathcal{I}_r \leq 1/r^2$ so $r^2\mathcal{I}_r \leq 1$).

Lemma D.4 combined with Lemma D.3 show condition 1 on Lemma D.2. Lemma D.3 shows condition 2. Lemma D.6 shows condition 3 for $k = 2$, and an essentially identical calculation shows condition 4 for $k = 2$. Lemma D.5 shows condition 3 for $k \geq 3$, and again an essentially identical calculation shows condition 4 for $k \geq 3$.

Thus, applying Lemma D.2 and Fact D.1, the failure probability of distinguishing $p = f_r$ and $q = f_r^{2\varepsilon}$ is at least

$$e^{-(1+O(\frac{\varepsilon}{r}\frac{1}{r^2\mathcal{I}_r})+O(1/\sqrt{n\varepsilon^2\mathcal{I}_r})+O(\varepsilon^2\mathcal{I}_r))n\varepsilon^2\mathcal{I}_r/2}$$

Picking

$$\varepsilon = \left(1 - O\left(\sqrt{\frac{\log\frac{1}{\delta}}{n}}\frac{1}{r^3\mathcal{I}_r^{1.5}}\right) - O\left(\frac{1}{\log\frac{1}{\delta}}\right) - O\left(\frac{\log\frac{1}{\delta}}{n}\right)\right)\sqrt{\frac{2\log\frac{1}{\delta}}{n\mathcal{I}_r}}$$

yields a failure probability lower bound of $\delta$, thus showing the theorem statement. $\qquad\square$