# OpenReview forum: "Finite-Sample Maximum Likelihood Estimation of Location"
_NeurIPS.cc/2022/Conference — NeurIPS 2022 Accept_

### Official Review · Reviewer_6bYV · 2022-07-09

**Rating:** 6
**Confidence:** 3
**Soundness:** 3 good
**Presentation:** 3 good
**Contribution:** 3 good

**Summary:**

This paper studies finite-sample location estimation: for a known distribution X with pdf f, given n samples from X+lambda, estimate lambda. In the large-sample limit n -> infty, it's known that maximum likelihood estimation is the best unbiased estimator with asymptotic variance 1/(n I), where I is the Fisher information of f. However, for a finite number of samples, there are pathological examples where the MLE variance is much larger than 1/(nI) until n hits some distribution-dependent threshold, and there are also examples where the MLE is suboptimal for small n.

This paper proposes smoothing each sample with independent additive Gaussian noise of some variance r^2, before applying MLE ("r-smoothed MLE"). The results are as follows:
1) r-smoothed MLE (with some additional algorithmic details) achieves, with probability 1-delta, deviations bounded by (1 + o(1))sqrt(2 log(1/delta) / (n I_r)), where I_r is the Fisher information of the smoothed distribution, where r = o(1) can be computed algorithmically
2) For sufficiently small delta and sufficiently large n, no algorithm given r-smoothed samples can perform location estimation with deviations bounded by (1-o(1))sqrt(2 log (1/delta) / (n I_r)) with probability 1-delta.
3) Experimentally, on a Laplace distribution with a Dirac delta spike, the smoothed MLE can outperform unsmoothed MLE and mean estimation algorithms when the number of samples n is small.


**Questions:**

1) The lower bound is only against algorithms that receive the smoothed samples. Thus, it gives no evidence that "smoothed MLE" is optimal when we are given the original samples. Indeed, given the pathological examples with Dirac delta spikes, it seems that an optimal algorithm may require examining the pdf in a more fundamental way?
2) The examples where the asymptotic behavior breaks down for finite-samples seem rather contrived. Certainly Dirac delta spikes may arise in practice (e.g. special 0s, or truncation) but I'm not sure why we would need location estimation in these settings. Is there some motivation for location estimation of pathological distributions where (a) regularity conditions of prior work fail, or (b) smoothing could be helpful?
3) It's noted that prior work on finite-sample bounds for MLE impose restrictive regularity conditions and lose multiplicative factors. I can't argue with the latter, but for the former I'm curious: does the smoothed PDF f_r not satisfy whatever regularity conditions are needed by the prior work, or if not what is the obstacle?
4) Is there some more explicit regularity condition under which the theory of this paper for the smoothed distribution f_r holds more generally (i.e. more explicit than the condition "this distribution is the convolution of some distribution with N(0,r^2)")?

**Limitations:**

Yes

**Strengths And Weaknesses:**


The paper studies an original problem (finite-sample location estimation) and is technically clear. As far as correctness, I did not go through all the proofs but the technical lemmas I checked seemed correct. However, the intended conceptual message is unclear to me. There are two issues at hand, which are both discussed in the introduction: (A) performing location estimation with near-optimal rates in the finite-sample regime; (B) predicting the performance of (any particular) algorithm (e.g. high-probability confidence intervals).

I think this paper makes an interesting theoretical contribution in Direction (B). Essentially, the theory in this paper shows that under a particular regularity condition (that the distribution can be obtained by convolving with a variance-r^2 Gaussian), the Fisher information governs the optimal estimation rates even with finite samples, and MLE obtains this optimal rate. If there is a way to estimate this smoothed Fisher information, then this seems to give a way to compute finite-sample confidence intervals. However, the theoretical contribution in direction (A) is more limited: it only shows that if the true distribution is already r-smoothed, then MLE is near-optimal in finite samples. For arbitrary distributions, the lower bound is not applicable.

The experiments are also somewhat limited; all they show is that for some pathological distributions, smoothed MLE can outperform MLE (which is already fairly clear from the paper's introduction). It's not immediately obvious to me why we would even care about location estimation for distributions with Dirac delta spikes (without knowing a priori where the spikes are). Also, there are no experiments validating that the theory leads to accurate confidence intervals for smoothed MLE.

I would weakly accept this paper on the strength of the insights that smoothing can (A) sometimes enable better location estimation (albeit demonstrated in pathological cases), and (B) allow for finite-sample error bounds. However, the paper would be significantly strengthened by (1) a clearer conceptual message supported by the theory and experiments, (2) lower bounds showing that (perhaps) smoothed MLE is optimal when the number of samples is finite, (3) more practically motivated distributions on which smoothing can be useful in practice, or (4) practical evidence that the theory provides tight confidence intervals.

---

> ### Author Response · Authors · 2022-08-02
> **Authors' response to Reviewer 6bYV**
>
> Thank you for your feedback and detailed comments. We agree that the paper has not completely resolved the algorithmic question (in the sense of (A) in the review), and we believe that the tools and theory developed in this work are a good step towards a complete understanding of the problem. The potential improvements you suggested are interesting directions for future work, which we are currently investigating.
>
> To answer your questions:
>
> 1. (Optimality) We agree that our lower bound does not actually show that smoothing is the optimal thing to do. It only shows that, conditioned on the smoothing, the MLE is optimal in the high probability regime. The problem of finding an actually optimal algorithm is an important direction that we are currently pursuing.
>
> 2. (Examples where prior work fails/smoothing is necessary)
> Without smoothing, the prior work doesn't apply even in simple cases like a standard Laplacian: [Spo11] because the KL divergence isn't locally quadratic in parameter distance, and [Pin17] both for this reason and because the score isn't smooth. We're not aware of any prior general theory that gets near-tight rates for the Laplacian.
>
>     So smoothing is clearly helpful as an analytical tool, letting us get a theory that applies to very reasonable distributions like the Laplacian.  But more than that, we can take an *arbitrary* distribution and extract a useful finite-sample bound from it.  The pathological examples show that the standard MLE does not achieve this, so smoothing is helpful algorithmically as well as analytically.
>
> 3. (Prior works) Below, we give a more detailed comparison with the cited papers. The main point is that none of the works apply well to Gaussian-smoothed distributions, even ignoring constant factors.  They do not support the phase transition in the sawtooth example, where with enough samples/low enough smoothing you can converge according to significantly higher Fisher information.
>
>     - [Mia10] requires Lipschitzness in the score, whereas for Gaussian-smoothed distribution, we only have 1-sided Lipschitzness (Lemma A.2). Even if the conditions hold, their results quantify the error not in terms of the Fisher information (the variance of the score), but in terms of the *maximum* magnitude of the score.
>
>     - [Spo11] is probably closest to our setting.  They require 6 regularity conditions, some of which are easy and some of which are not obvious but hold (e.g., our Corollary 3.3 shows that their condition (eu) on Page 21 holds for $f_r$).  But one of their conditions, (lu), does not hold.  It assumes that the parametric family $\{f^{\theta}\}$ satisfies $D_{KL}(f^\theta  ||  f^{\theta'}) \ge \Omega(I |\theta - \theta'|^2)$. This is a *global* distance property that (in examples like the sawtooth) does not always hold.  This is why our final algorithm (Algorithm 2) has two stages.
>
>     - [Pin17] requires a similar global distance property to [Spo11], but in squared Hellinger distance rather than KL divergence.  (These are conditions (D0), (D1).)
>
> 4. (Generalizing the theory, regularity conditions) Our proof of Theorem 1.1 only uses the following properties of Gaussian-smoothed distributions: 1) the score function is well-concentrated for each point in a small neighborhood around the true parameter (Corollary 3.3), 2) at points around the true parameter, the score function has well-controlled mean and second moment (Lemmas B.2 and B.3), and 3) the score function always satisfies $s'(x) \ge -s_0$ for some constant $s_0 \ge 0$ (Lemma A.2).

---

> > ### Comment · Reviewer_6bYV · 2022-08-07
> > **Thanks**
> >
> > Thanks for answering my questions. The comparison with prior work seems helpful to include in the text. I maintain my original evaluation.

---

### Official Review · Reviewer_64zQ · 2022-07-10

**Rating:** 6
**Confidence:** 4
**Soundness:** 3 good
**Presentation:** 3 good
**Contribution:** 3 good

**Summary:**

The paper explore the limits of the Maximum likelihood estimator (MLE) when the density is not smooth. Usually, the Fisher information gives the optimal deviations for the MLE, however there are a number of cases where the Fisher information is infinite or very large and it can't be used anymore to derive deviation bounds. In such cases, the authors propose to use the Fisher information of a smoothed version of $f$ instead. Using this, the authors give among other things a way to construct a robust location estimator for a given (non smooth) distribution.

**Questions:**

- My main concern is the choice of the optimal value of $r^*$ to apply the results when one only wants to estimate the location parameter. In Theorem 5.1, a loose lower bound is given for $r^*$ even though $I_{r^*}$ can change a lot with $r^*$. A wrong choice of $r^*$ would entail an unneeded multiplicative error that renders the $(1+o(1))$ guarantees useless. This comment also applies to Theorem 1.3. Could the authors comment on that ?

- Is it possible to quantify what we loose by considering $I_{r}$ instead of $I$ ? For instance, if the data come from a distribution that is smooth, is the value of $I_r$ very different from $I$ ?

- The authors talk about the robust context, how the article could be used in the case of Gaussian with a Dirac spike. Would it be possible to state an estimator and theoretical guarantees that suppose only knowledge of some inlier distribution and allow for an unknown outlier distribution that render the problem non-smooth ? To compare to state of the art results in robust theory.

- In the experimental results of Section 7, the authors use a simplified version of the Algorithm proved in practice. Would the algorithm originally proposed work in practice ? How does the efficiency compare between the algorithm used in Section 7 and Algorithm 1 ?

- Could the authors include the median in the experiments ? The median is known to handle non-smoothness when estimating location parameters. It would be interesting to compare with the proposed estimators.

- Could the authors explain Lemma 3.2 ? This Lemma does not seem to be true when $r$ goes to $0$. Could the authors detail the coupling argument used here ?


**Limitations:**

Suggestion of improvements are included in the questions. In particular, the practical applications and  motivations should be developed a bit more.

**Strengths And Weaknesses:**

Strengths: the paper is clear, the proofs are clear. Some of the results and proof techniques are of independent interest. The guarentees are given with high probability for results that usually hold only asymptotically.

Weaknesses: the paper is poorly motivated, there are some mention of examples in which the technique could be useful but these examples are not developed enough to compare to the state of the art.

---

> ### Author Response · Authors · 2022-08-02
> **Authors' response to Reviewer 64zQ**
>
> Thank you for your questions and suggestions. They are helpful to us for refining the message in the paper. We address your questions below.
>
> 1. (Choice of $r^*$) Given the guarantees of Theorem 5.1, we should always choose the smallest value of $r^*$ such that the theorem holds. To be explicit, in the statement of Theorem 5.1, when we write "Choose $r^* = \Omega(\text{expression})$", we mean that there exists a universal constant $c$ such that if we choose any $r^* \ge c \cdot \text{expression}$, then the theorem holds. To use the result, we would then choose $r^*$ equal to exactly $c \cdot \text{expression}$.
>
>     As for Theorem 1.3, the statement assumes that the underlying distribution has already been smoothed. The theorem then shows that the MLE performs optimally, conditioned on this smoothing.
>
> 2. (Comparison of $I_r$ to $I$ under smoothness assumptions)
> $I$ characterizes the limiting convergence, while we show that $I_r$ upper bounds the finite-sample convergence.  These can be arbitrarily far apart, even under smoothness assumptions (e.g., the first $O(1)$ derivatives are $O(1)$), so $I$ and $I_r$ can also be very far apart.
>
>     In particular, one can construct a function with $O(1)$ derivatives being $O(1)$ but arbitrarily large Fisher information [one example: $A \sin^2(Fx)\exp(-x^2/R)$ for appropriate $A, F, R$].  Mixing this with a Gaussian leads to the above claim (we can include a more formal proof in the paper if you think it would be helpful).
>
> 3. (Robustness) In this work, we do not consider any robustness in the sense of distributional (Huber) contamination. It is certainly a very interesting direction, since one of the main downsides to parametric estimation theory is the assumption of complete knowledge of the model, which the robust version of the problem addresses. Huber's seminal work [Hu64] demonstrated that, if an adversary can choose an arbitrary distribution from a class $\mathcal{F}$ that is sufficiently regular, the best possible asymptotic variance for any location estimator is the reciprocal of the *smallest* Fisher information over the class $\mathcal{F}$. This asymptotic variance is attained by the MLE which assumes the distribution $F_0 \in \mathcal{F}$ that has the smallest Fisher information. It is an interesting question to see if Huber's "minimax" theory can be generalized to finite-sample and high-probability setting via Gaussian smoothing.
>
> 4. (Proposed vs simplified algorithm) The proposed algorithm should also work well. The simplification in the algorithm involve 1) examining the entire real line as opposed to some interval known to contain the true parameter, and 2) performing actual maximum likelihood finding instead of root finding on the empirical score function. Both simplifications (potentially) decrease performance.
>
> 5. (Performance of median) In the particular example of the spiked Laplace model, the median does perform well, since the median is the MLE of the Laplace distribution. In fact, the median performs essentially the same as the smoothed MLE. On the other hand, there are many other distributions where the MLE is *not* the median; even for a Gaussian the median is inefficient.
>
> 6. (Lemma 3.2, proof) Lemma 3.2 is true at $r = 0$, according to the citation [SV11]. To prove the lemma in general, as we explained, it suffices to show the following: Given a distribution $f$ with interquartile range $IQR$, and denote the 30th-70th percentile range of $f_r$ as $R$, then $R \le IQR+O(r)$.
>
>     Here is a simple argument for the above claim. Let $q_\ell$ be the 25th percentile of $f$. Drawing a sample from $f_r$ is equivalent to independently drawing $x$ from $f$ and $z$ from $N(0,r^2)$ and returning $x+z$. With probability at least 0.75, $x \ge q_\ell$. Also, with probability at least 0.95, $z \ge -\Theta(r)$. Therefore, by a union bound, we have $x+z \ge q_\ell - \Theta(r)$ except with probability at most 0.3, meaning that the 30th percentile of $f_r$ is at least $q_\ell - \Theta(r)$. Combined with the symmetric argument for the 70th percentile of $f_r$, this shows that $R \le IQR + O(r)$.

---

### Official Review · Reviewer_dJ3A · 2022-07-12

**Rating:** 7
**Confidence:** 4
**Soundness:** 3 good
**Presentation:** 2 fair
**Contribution:** 3 good

**Summary:**

This paper studies a classical object in statistics, the maximum likelihood estimator (MLE), in estimating the location parameter in a location family. Although the inverse Fisher information completely characterizes the optimal estimation error asymptotically, this estimation error may not be achieved in finite samples. In this paper, the authors proposed to convolute the target distribution using another Gaussian, in order to make the distribution smoother and better behaved. The benefits of doing so are two-fold:

1. The estimation error of MLE in the convoluted model approximately achieves the inverse Fisher information, even in finite samples.

2. The above error is also near-optimal by a complementing (two-point) minimax lower bound, which holds in finite samples as well.

Experimental results are also included to illustrate the performance of the proposed estimator.

**Questions:**

I am looking forward to the authors' responses to my above points.

**Limitations:**

The authors have adequately discussed the limitations in the future direction section.

**Strengths And Weaknesses:**

This paper provides an interesting finite-sample view of the MLE, i.e. when the family of underlying distribution is very smooth, the asymptotic performance of MLE also holds in finite samples. A key feature in the result is that the authors aim to get the optimal constant, instead of a rate-optimal analysis. This paper is technically solid, and the results are sound.

Despite my appreciation on the technical front, I am not sure of the usefulness of both the theory and methodology:

1. On the methodology side, I am not convinced by why people should choose the smoothed MLE over the classical one. There is no formal theorem showing the failure of the MLE in finite samples, and since the smoothed Fisher information is smaller than the original one, a practitioner is actually paying a price on the variance for the guarantee on the finite sample result. Also, the authors did not comment on the computational aspect of the smoothed MLE, as the likelihood of the convolution becomes more complicated. I am wondering if the purpose of this paper is really to advocate the usage of the smoothed MLE in practice.

2. On the theory side, even if this should be purely treated as a theoretical paper, the authors should make more comparisons of their results and treatments with the statistical literature. The finite sample analysis of the MLE is not new; there is a well-known result using the empirical process theory which could upper bound the finite-sample performance of the MLE in terms of the Dudley integral of the Hellinger metric entropy. A nice reference is the following book:

Sara A van de Geer. Empirical Processes in M-estimation, volume 6. Cambridge university press, 2000.

This line of research seems to be overlooked in this paper. Although the above only provides rate analysis, it would be great to see the connections. For example, when the location family formed by Gaussian convolution, would the empirical process theory lead to similar finite sample results in terms of the Fisher information?

Other comments:

If the authors are mainly investigating theories of MLE, I would actually suggest the authors to tell the story in another way, i.e. establish a nice finite-sample result for the MLE when the distribution is smooth (i.e. a Gaussian convolution), rather than advocating an artificial convolution of another Gaussian distribution. I think this is still a nice storyline of the theory while avoids controversies on the practical front.

Lemma 3.1 is a direct consequence of the following inequality on the Fisher information by Stam: for independent X and Y, 1/I(X+Y) >= 1/I(X) + 1/I(Y).

Lemma C.2: conditions 5 and 6 cover conditions 3 and 4 by choosing k = 2.

Post rebuttal: Thanks to the authors for the clarification and I am willing to increase my score to 7.

---

> ### Author Response · Authors · 2022-08-02
> **Authors' response to Reviewer dJ3A**
>
> Thank you for your comments and suggestions. We address your concerns below.
>
> - (Failure of MLE) We gave formal hardness statements for these examples in our top-level response, do they address this question?
>
> - (Computational aspect of smoothed MLE) When a distribution is Gaussian-smoothed, then it has bounded derivatives, and it is safe to discretize it for the purposes of computing the MLE. In order to compute the smoothed version of the distribution, all we have to do is to discretize the original distribution and convolve it with the (discrete) Gaussian. Since the setting is currently 1-dimensional, such discrete convolution is relatively cheap to compute.
>
> - (Other related works) Thank you for pointing out literature that we missed. We are happy to compare with any additional missed references as well.
>
>     The van de Geer book shows convergence of the recovered distribution in Hellinger distance to the true distribution, at a rate dependent on a metric entropy bound (and lossy of constant factors).  To apply this in our setting for Gaussian-smoothed distributions, we need (I) to convert this to a bound on parameter distance, and (II) to relate the metric entropy to Fisher information.  We believe that both these are possible to do in a local neighborhood around the true parameter (though this is not easy), but the metric entropy needs to be bounded at global distances as well.  This seems unlikely to hold in our setting.
>
>     This issue is closely related to the assumptions made in [Spo11] and [Pin17], where they explicitly assume KL-divergence and Hellinger distance lower bounds in terms of parameter distance. Please see our response to Reviewer 6bYV for more details.
>
> - (Misc comments on the lemmas) Thank you for your careful reading. We will fix them.

---

### Official Review · Reviewer_hCp9 · 2022-07-12

**Rating:** 6
**Confidence:** 3
**Soundness:** 2 fair
**Presentation:** 2 fair
**Contribution:** 2 fair

**Summary:**

The paper proposes a maximum likelihood estimation technique that gives a non-asymptotic recovery guarantee of the location parameter $\lambda$ of the underlying distribution of the generated samples. The density model $f$ is assumed to be translation invariant. It is known that MLE estimated parameters converge to their true value in a limit where a number of samples n tends to infinity. MLE also achieves the Cramer-Rao lower bound of the variance which is $1/nI$ where I is the Fisher information of $f$ in the limit. With access to a finite number of samples, this paper claims to attain with a high probability the variance lower bound within (1+o(1)) factor.

The work is quite relevant when the algorithm is not given access to an infinite number of samples, which can very well occur in real-world scenarios, where the system demands successful predictions with a high probability guarantee.

The key technique to attain the claim made by the authors is to simulate a Gaussian smooth version of the model by adding Gaussian noise $N(0,r^2)$ to each point and performing MLE corresponding to the r-smooth version of model $f$. The smoothing parameter r, which is the stdev of the added Gaussian noise, is calculated based on the targetted optimality margin. Doing MLE on the model is equivalent to finding the roots of the corresponding empirical score function $s(\hat{\lambda})$ as given by the standard definition of the score.

The paper has presented two examples of distributions where taking a finite number of samples and performing MLE leads to a high variance estimate.

1) In the first example, Gaussian with sawtooth noise, it is not clearly explained how taking $n>>1/w^2$ samples ‘aligns’ the teeth and how the variance $\Omega(1)/(\Delta^2 n)$ is obtained. Also, how does taking smaller $n$ gives a lower variance $1/n$

2) It says in line no. 57 that because of the sawtooth gradient, Fisher information $I$ increases. Does it not mean that the variance, being inverse of $I$, decreases?

3) In the second example: Gaussian with Dirac $\delta$ spike, it is not clear why the case where we don't see a point from the spike due to $n<1/\epsilon$ is not better because in the absence of an outlier point MLE is privileged to recover $\mu$ better. How does seeing a point from the spike help in estimating location?

4) The effect of adding noise to better estimate location in the second example in lines no 76-78 is not elaborate and clear enough.

The authors have presented a high probability optimal recovery guarantee of their algorithm in Theorem 1.1 and 1.2. Algorithm 1 represents Local MLE where the uncertainty region of the location parameter is provided to the algorithm. Algorithm 2 represents two-stage global MLE where the first stage is to estimate the uncertainty region from the interquartile range(IQR) of the model $f$, and the second stage is to run algorithm 1 with the obtained range.

5) In theorem 1.1, taking $\gamma$ to be ‘sufficiently large’ goes against its use in the three inequalities in line no. 99.

**Questions:**

Please see above.

**Limitations:**

Yes.

**Strengths And Weaknesses:**

Please see above.

---

> ### Author Response · Authors · 2022-08-02
> **Authors' response to Reviewer hCp9**
>
> Thank you for your feedback and questions. Below, we address the questions/points raised in the review:
>
> 1. (Needing enough samples to utilize sawtooth)  We will include a formal statement of this result (see the general response).  Since the Fisher information is $\Delta^2$, when $n$ is sufficiently large the variance should be $O(1/(\Delta^2 n))$.
> When $n \gg 1/w^2$, the basic variance bound (of $1/n$, which might not be tight)  shows that we will estimate to within the correct "tooth"; this is where the phase transition happens to converge according to the Fisher information. This allows us to conclude that the true variance is in fact $O(1/(\Delta^2 n))$, which is much smaller than the basic variance bound.
>
> 2. (Sawtooth increases Fisher information)
> The Fisher information is the variance of the score.  The sawtooth gradient makes the score quite large (alternating positive and negative), so it increases the Fisher information.
>
> 3. (Spiked Gaussian is hard for MLE) The fact that not seeing an outlier would ruin the MLE is indeed counterintuitive at first. Recall that the Dirac $\delta$ spike has *infinite* density (even if it only has very little total mass). Therefore, given a set of samples, the MLE necessarily returns an estimate such that (at least) one of the samples is fitted to the spike: this estimate has infinite empirical likelihood, whereas any estimate that does not fit any sample to the spike has finite empirical likelihood which is strictly smaller. If the set of samples never contained any observations from the spike, and if the spike is very far from the true mean, then a large estimation error is incurred.
>
> 4. (Smoothing improves MLE for spiked Gaussians) The issue in 3 is that a Dirac $\delta$ spike has infinite density, which always makes the MLE fit a sample to the spike. Convolving a Dirac $\delta$ with a Gaussian reduces the density from infinity down to the density of the Gaussian itself.  The MLE will no longer overfit to the spike.
>
> 5. (Theorem 1.1 phrasing) By "sufficiently large" we mean that there exists universal constant such that Theorem 1.1 holds whenever $\gamma$ is larger than this constant. In particular, the constant 100 suffices (and we have not looked into the proofs to optimize this constant yet).  Subject to this requirement, smaller $\gamma$ makes the three required inequalities more likely to hold, but also makes the conclusion weaker.

---

### Author Response · Authors · 2022-08-02
**Overall response**

Thank you for your in-depth comments and feedback. We will address the individual questions and comments directly under each of the reviews. Given that multiple reviewers asked questions about the hard examples we discussed in Section 1.1, here we give formal statements of their implications.

For the Gaussian+sawtooth example, we have the following information-theoretic lower bound theorem, lower bounding the error of any estimator. For simplicity in notation, we give the constant probability result.

---

Theorem [Gaussian+Sawtooth]

The Gaussian+sawtooth model, with "teeth" of width $w$ and slope $\Delta$, has Fisher information $\Delta^2 \gg 1$ but no location estimator can have error $o(1/\sqrt{n})$ with constant probability over $n$ samples, unless $n > 0.01/w^2$.

This holds for arbitrarily large $\Delta$ and small $w$.  By contrast, the asymptotic theory predicts error $\Theta(\frac{1}{\Delta \sqrt{n}})$, which only holds for $n \gtrsim 1/w^2$.

---

Proof sketch

Let $f$ denote the Gaussian with sawtooth model, and $f^{(\epsilon)}$ denote the model but shifted by a distance of $\lceil \epsilon/w \rceil w$ (that is, the largest number of sawteeth that fits into a distance of $\epsilon$).

Then, the KL divergence satisfies $D_{KL}(f, f^{(\epsilon)}) = O(\epsilon^2)$. By Pinsker's inequality, this implies that we need $\Omega(1/\epsilon^2)$ samples to distinguish $f$ and $f^{(\epsilon)}$ with constant probability. For $\epsilon > w$, the shift between $f$ and $f^{(\epsilon)}$ is at least $\epsilon/2$.

Concluding, if $n < 1/(100w^2)$, then we set $\epsilon = 1/(10\sqrt{n})$ and there is no algorithm that can distinguish $f$ and $f^{(1/(10\sqrt{n}))}$ using $n$ samples with constant probability, meaning that no location estimator can achieve error $1/(20\sqrt{n})$ with constant probability. The reasoning generalizes to the high probability regime as well.

---

For the spiked Gaussian example, the reasoning in Section 1.1 implies both an information-theoretic lower bound, and also a lower bound of the error of the (unsmoothed) MLE.

---

Theorem [Spiked Gaussian]

Consider the spiked Gaussian example $(1-\tau) N(0, 1) + \tau \delta_T$ where $\delta_T$ is a Dirac $\delta$ at location $T$.

The Fisher information is infinite, but for $n < \frac{1}{100 \tau}$ every estimation algorithm has median error at least $\frac{0.6}{\sqrt{n}}$.  Moreover, with $98$% probability the MLE has error at least $T - O(\sqrt{\log n})$, which can be arbitrarily large.

---

It is also worth pointing out (as we do in our response to Reviewer 6bYV) that, while we present a hardness result using a Dirac $\delta$ spike for simplicity, a milder but similar effect occurs even when the spike is narrow but not infinite.

---

### Meta-Review · Area_Chair_VW8w · 2022-08-26

**Recommendation:** Accept
**Confidence:** Less certain

**Metareview:**

This paper considers the problem of parameter estimation using smoothed observations: indeed, after smoothing although the asymptotic variance may increase we also have finite sample guarantees for the MLE. Although it is not clear how useful this methodology would be for high-dimensional applications, I am impressed by this neat observation and the solid writing.

**Award:**

No

---

### Decision · Program_Chairs · 2022-09-14

Accept